# Epsin deficiency impairs endocytosis by stalling the actin-dependent invagination of endocytic clathrin-coated pits

**Mirko Messa[1], Rubén Fernández-Busnadiego[1†], Elizabeth Wen Sun[1], Hong Chen[1‡], Heather Czapla[1], Kristie Wrasman[2], Yumei Wu[1], Genevieve Ko[1], Theodora Ross[3], Beverly Wendland[2], Pietro De Camilli[1]\***

[1]Program in Cellular Neuroscience, Neurodegeneration and Repair, Department of Cell Biology, Howard Hughes Medical Institute, Yale University School of Medicine, New Haven, United States; [2]Department of Biology, Johns Hopkins University, Baltimore, United States; [3]Department of Internal Medicine, UT Southwestern Medical Center, Dallas, United States

**Abstract** Epsin is an evolutionarily conserved endocytic clathrin adaptor whose most critical function(s) in clathrin coat dynamics remain(s) elusive. To elucidate such function(s), we generated embryonic fibroblasts from conditional epsin triple KO mice. Triple KO cells displayed a dramatic cell division defect. Additionally, a robust impairment in clathrin-mediated endocytosis was observed, with an accumulation of early and U-shaped pits. This defect correlated with a perturbation of the coupling between the clathrin coat and the actin cytoskeleton, which we confirmed in a cell-free assay of endocytosis. Our results indicate that a key evolutionary conserved function of epsin, in addition to other roles that include, as we show here, a low affinity interaction with SNAREs, is to help generate the force that leads to invagination and then fission of clathrin-coated pits.

**\*For correspondence:** pietro. decamilli@yale.edu

**Present address:** [†]Max-Planck-Institut für Biochemie, Martinsried, Germany; [‡]Cardiovascular Biology Research Program, Oklahoma Medical Research Foundation, Oklahoma City, United States

**Competing interests:** The authors declare that no competing interests exist.

## Introduction

Clathrin-mediated endocytosis involves a complex set of factors besides clathrin itself and the classical clathrin adaptors. These factors help coordinate nucleation of the clathrin coat with cargo selection, membrane invagination and fission (*Schmid, 1997*; *Slepnev and De Camilli, 2000*; *Kaksonen et al., 2003*; *Merrifield et al., 2005*; *Ferguson and De Camilli, 2012*). An important player in these processes is epsin, the collective name for a family of evolutionarily conserved clathrin-associated proteins (*Chen et al., 1998*; *Wendland, 1999, 2002*; *Ford et al., 2002*; *Chen and De Camilli, 2005*), which in mammals is represented by three isoforms: epsin 1, 2, and 3 (encoded by the *Epn1*, *Epn2*, and *Epn3* genes, respectively [*Ko et al., 2010*]). Epsin was identified as a major interactor of Eps15 (*Chen et al., 1998*), another clathrin coat associated protein. It comprises a membrane binding N-terminal ENTH (Epsin N-Terminal Homology) domain, which is followed by ubiquitin-interacting motifs (UIMs [*Polo et al., 2002*]) and a long sequence (tail) predicted to be primarily unfolded and flexible (*Wendland, 2002*). The core of the ENTH domain is preceded by a short sequence that is unfolded in solution but folds into an amphipathic α-helix upon binding to PI(4,5)P$_2$. The hydrophobic portion of the helix partially penetrates the bilayer, thus conferring membrane curvature generation and sensing properties to the protein (*Itoh et al., 2001*; *Ford et al., 2002*). Epsin's disordered tail binds components of the clathrin coat via multiple short amino acid motifs: 'clathrin boxes' bind clathrin, DPW/F motifs bind the appendage domain of AP-2, and NPF motifs bind the EH domains of Eps15 and intersectin (*Chen et al., 1998*; *Rosenthal et al., 1999*; *Drake, 2000*; *Shih et al., 2002*; *Overstreet et al., 2003*).

**eLife digest** Clathrin-dependent endocytosis is one of the mechanisms used by cells to internalize specific proteins (cargo) from their surface. First, the cargo interacts with adaptor proteins that help cluster them in the cell's outer membrane, called the plasma membrane. This causes the protein clathrin to assemble into a lattice at the cytosolic side of the plasma membrane and deform the membrane into a pit. The pit grows deeper over time as more clathrin molecules assemble, eventually resulting in a deeply invaginated clathrin-coated pit that encloses the cargo to be taken up by the cell. The clathrin-coated pit then pinches off inside the cell in a process called fission to form a bubble-like structure called a vesicle, which transports the molecule to its destination.

The deep invagination of clathrin-coated pits that leads to fission is assisted by actin, a protein that assembles into filaments that are suggested to generate the forces needed for this process. Many other factors are also involved. One of them is epsin, the collective name for a family of three very similar proteins in mammalian cells. Epsin binds to several other proteins implicated in clathrin-dependent endocytosis, including clathrin itself, and to plasma membrane proteins specifically 'tagged' for internalization. In addition, a portion of the epsin molecule can insert into the plasma membrane and help it to curve, which is important for forming the invaginated pit. However, due to the number of possible functions epsin could perform, its main role has remained elusive.

Messa et al. created mouse cells that lack all three epsin proteins. Although these cells can form clathrin-coated pits, they struggle to develop into vesicles. The normal linking of the actin filaments to the clathrin coat does not occur, and another protein called Hip1R that also participates in clathrin-mediated endocytosis and links clathrin to actin, no longer accumulates at the clathrin-coated pits. Messa et al. also find that epsins can bind directly to actin. Overall, these results suggest that a main role of epsin is to help actin interact with the clathrin-coated pits and generate the force required for a pit to develop into a vesicle. However, epsin also performs many other roles, including recruiting a membrane protein (a so-called SNARE) that directs the fate of the vesicle to the clathrin-coated pit.

Additionally, Messa et al. find that cells lacking all three epsins have problems dividing correctly. More research is required to establish whether this effect is also due to epsin's interaction with the cell's actin cytoskeleton.

As epsin binds ubiquitin and genetically interacts with enzymes of ubiquitin metabolism (*Cadavid et al., 2000*; *Chen et al., 2002*; *Polo et al., 2002*; *Shih et al., 2002*; *Chen et al., 2003*; *Sigismund et al., 2005*), it was proposed to function as a clathrin adaptor for ubiquitinated cargo. Strong evidence for such a role came from the demonstration of Notch signaling defects in epsin (*liquid facets*) mutant flies, as Notch signaling is critically dependent upon ubiquitin-dependent endocytosis of Notch ligands (*Overstreet et al., 2003*; *Xie et al., 2012*). However, other findings pointed to a general house-keeping role of epsin in clathrin-mediated endocytosis. Absence of the two epsins (*Ent1* and *Ent2*) in yeast is lethal, while hypomorphic *Ent1* or *Ent2* mutations result in defects in endocytosis and actin dynamics (*Wendland, 1999*; *Aguilar et al., 2003*; *Skruzny et al., 2012*). Impairments in clathrin and actin function were also observed in epsin null *Dictyostelium* mutants (*Brady et al., 2008*; *2010*). In both these unicellular organisms, epsin functions in close cooperation with Sla2/Hip1R, another evolutionarily conserved clathrin accessory factor (*Brady et al., 2008*; *2010*; *Skruzny et al., 2012*). However, a link between epsin and Hip1R in metazoan cells has not been reported.

Hip1 family members (Hip1 and Hip1R in mammals) comprise an N-terminal ANTH domain followed by unfolded regions that bracket a coiled-coil region and a C-terminal THATCH (talin-HIP1/R/Sla2p actin-tethering C-terminal homology) domain (*Engqvist-Goldstein et al., 1999*; *Wilbur et al., 2008*; *Skruzny et al., 2012*). The coiled-coil region can homo-heterodimerize and also binds clathrin light chain (*Engqvist-Goldstein et al., 2001*; *Metzler et al., 2001*; *Legendre-Guillemin et al., 2002*; *Gottfried et al., 2010*). The THATCH domain is an actin-binding module (*Yang et al., 1999*; *Engqvist-Goldstein et al., 2001*; *Brett et al., 2006*; *Wilbur et al., 2008*). Accordingly, Sla2/Hip1R binds actin and is thought to function as a major link between the clathrin coat and actin. Studies in yeast have additionally shown that the ENTH domain of epsin and the ANTH domain of Sla2 interact with each other, and the two proteins function together in providing a link between the endocytic coat and the actin cytoskeleton (*Skruzny et al., 2012*).

In addition to roles of epsin mediated by protein–protein interactions, membrane remodeling properties resulting from the amphipathic helix at the N-terminus of its ENTH domain have been implicated in the clathrin-dependent endocytic reaction. In vitro studies showed that this helix confers, upon the ENTH domain, the property to induce bilayer curvature and even to fragment bilayer tubules into vesicles, thus pointing to a potential role of the epsin in fission (*Itoh et al., 2001*; *Ford et al., 2002*; *Boucrot et al., 2012*).

Surprisingly, in view of this evidence for an important housekeeping role of epsin in endocytosis, the germline knockout (KO) of the mouse *Epn1* and *Epn2* genes that encode the two major ubiquitously expressed mammalian epsins, epsin 1 and 2, did not block the early embryonic development (*Chen et al., 2009*). Arrest of embryonic development occurred only at E9.5–E10, with a pattern suggestive of impaired Notch signaling, while no obvious defects in clathrin-mediated endocytosis were observed in fibroblasts derived from these embryos (*Chen et al., 2009*). Moreover, studies of epsin 1 and 2 conditional double KO endothelial cells revealed a selective defect in the internalization of ubiquitinated VEGF receptor (*Pasula et al., 2012*). However, a recent study based on RNAi-mediated knock-down (KD) in fibroblastic cells reported that the KD of all the three epsins produces a global impairment of clathrin-mediated endocytosis, which was attributed to a defect of the fission reaction (*Boucrot et al., 2012*).

The goal of the present study was to provide conclusive evidence about the function(s) and sites of action of epsin in endocytosis using a gene KO strategy to completely eliminate all epsins. Our results, which capitalize on triple KO (TKO) cells generated from conditional epsin TKO mice, show that epsin provides a link between the clathrin coat and actin and is needed for the transition of pits from a shallow to a deeply invaginated state. As in unicellular organisms, epsin acts in concert with Hip1R but with differences from yeast to mammals. An additional function of epsin is a low affinity interaction of its ENTH domain with synaptobrevin 2/VAMP2 (Syb2) that may help ensure the presence of a vesicular SNARE in the budding vesicle.

## Results

### Generation of conditional epsin triple knockout cells

As the germline deletion of even only two *Epn* genes results in embryonic lethality, a conditional approach was used to generate *Epn1*, *Epn2*, and *Epn3* triple KO cells. Towards this goal, *Epn1*$^{loxP/loxP}$ (*Figure 1—figure supplement 1A*, see also [*Pasula et al., 2012*]) mice were crossed with *Epn2* (*Chen et al., 2009*) and *Epn3* KO (*Ko et al., 2010*) mice to generate *Epn1*$^{loxP/loxP}$; *Epn2*$^{-/-}$; *Epn3*$^{-/-}$ animals, which were viable and fertile with no obvious pathological phenotypes. These mice were subsequently interbred with mice transgenic for 4-hydroxy-tamoxifen (OHT)-inducible Cre recombinase [Cre-ER, *Badea et al., 2003*] to obtain *Epn1*$^{loxP/loxP}$; *Epn2*$^{-/-}$; *Epn3*$^{-/-}$; Cre-ER$^{+/0}$ animals. These mice did not exhibit obvious defects either. Conditional epsin TKO mouse embryonic fibroblasts (MEFs) were derived from these animals via treatment with OHT, whose action was confirmed by the translocation of Cre into the nucleus (*Figure 1—figure supplement 1B*), and typically examined after 7 days of OHT treatment (after 9 days of treatment cell death started to occur). Treated MEFs derived from the same litters but harboring a wild-type (WT) epsin 1 allele did not exhibit any obvious differences from OHT-treated WT fibroblasts in terms of proliferation, endocytosis and actin organization. In the experiments described below OHT-treated WT MEFs were used as controls.

Immunoblotting with epsin isoform-specific antibodies demonstrated the near complete disappearance of epsin 1 from conditional TKO cell extracts in response to OHT (*Figure 1A*) and confirmed the absence of epsin 2 and 3 (*Figure 1B*). The extremely small amount of residual epsin 1 was likely explained by delayed gene recombination in a few cells, where residual epsin 1 immunoreactivity, much lower than in controls, was observed by immunofluorescence. A C-terminal epsin 2 fragment (MW about 40 kDa) was detected in cells containing the epsin 2 KO allele, possibly reflecting an alternative start site (*Figure 1—figure supplement 1C*). However, the KD of this fragment by RNAi in TKO cells did not produce phenotypic changes in addition to the ones described below, indicating that it does not play a relevant role. OHT-dependent loss of epsin 1 expression was also validated by the anti-epsin 1 immunofluorescence, as the typical punctate epsin 1 signal (reflecting clathrin-coated pits [*Chen et al., 1998*]) was completely absent (*Figure 1C*).

We also mated mice with mutations in all the three *Epn* genes with nestin-Cre transgenic mice (*Tronche et al., 1999*) to obtain brain-specific TKO mice (genotype: *Epn1*$^{loxP/loxP}$; *Epn2*$^{-/-}$; *Epn3*$^{-/-}$;

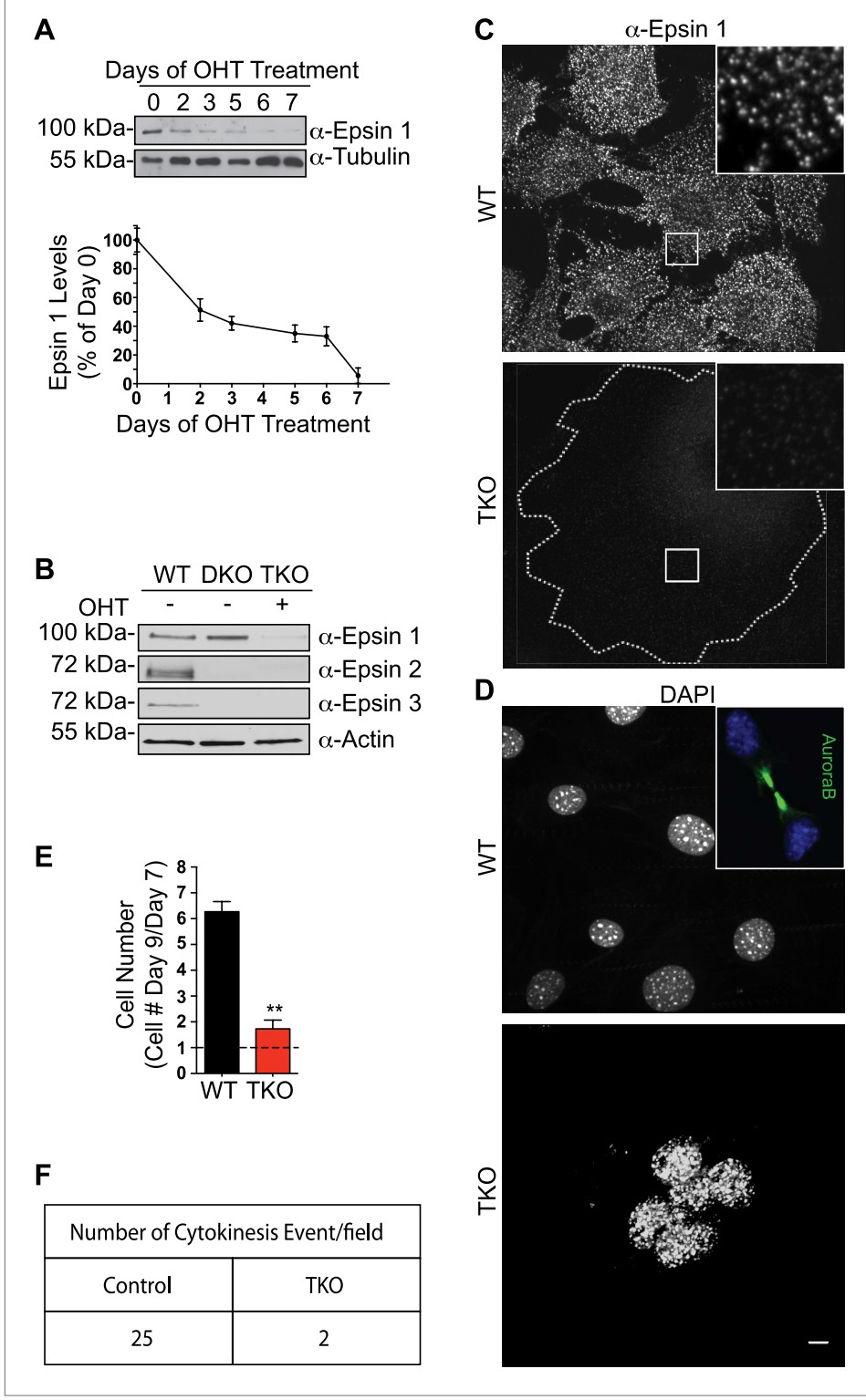

**Figure 1**. Mitotic defects in epsin TKO fibroblasts. (**A**) Anti-epsin 1 immunoblot shows the disappearance of epsin 1 from *Epn1*<sup>loxP/loxP</sup>; *Epn2*<sup>−/−</sup>; *Epn3*<sup>−/−</sup>; Cre-ER<sup>+/0</sup> cells after 7-day treatment with 4-hydroxy-tamoxifen (OHT). Tubulin was used as a loading control (top). Densitometric analysis of the epsin 1 band during OHT-treatment (bottom). (**B**) Anti-epsin immunoblots with isoform-specific antibodies showing OHT-treatment for 7 days results in the loss of epsin 1 in epsin 2/3 double knock-out (DKO) thus generating triple KO (TKO) cells. Wild type (WT) cell lysate was
*Figure 1. Continued on next page*

*Figure 1. Continued*

used as a control. (**C**) Anti-epsin 1 immunofluorescence shows that the typical punctate epsin 1 signal of WT cells (top) was completely absent in TKO cells (bottom). The perimeter of the TKO cell is indicated by a dotted white line and demonstrates a very large size relative to WT cells. The insets show higher magnification of the boxed regions. (**D**) DAPI staining showing single nuclei in WT and multiple nuclei in a TKO cell. The inset of the WT field shows the accumulation of AuroraB kinase immunoreactivity at the mid-body during cytokinesis. (**E**) The increase in cell number during a 3-day incubation is lower in TKO cells. Cells were counted at day 7 and 9 after addition of OHT (**$p < 0.01$, Student's *t* test, n = 3 experiments). (**F**) As shown by a morphometric analysis, cytokinesis events are only rarely observed in TKO cells. Scale bar represents 10 µm. Data are represented as mean ± SEM. See also *Figure 1—figure supplement 1*.

The following figure supplement is available for figure 1:

**Figure supplement 1**. The generation of triple KO cells, nestin Cre brain specific triple KO mice, and the observation of nuclear defect in TKO cells.

Nestin-Cre$^{+/0}$) and their controls with a WT epsin 1 allele (*Epn1*$^{loxP/+}$; *Epn2*$^{-/-}$; *Epn3*$^{-/-}$; Nestin-CRE$^{+/0}$). Nestin-Cre epsin TKO animals were born, but at a lower than expected Mendelian ratio. A few days after birth they began to develop locomotor dysfunction, failed to gain weight (*Figure 1—figure supplement 1D*), and rapidly deteriorated, typically dying before the end of the fourth week. Western blot analysis of extracts of their brains at the beginning of the fourth week, confirmed the absence of all the three epsin genes (*Figure 1—figure supplement 1E*).

In this study, we focused on the impact of the lack of epsin on basic cellular properties using MEFs and cell-free assays. Brain specific epsin TKO mice will be characterized in a future study, but we made use of their brains as the source of epsin TKO brain cytosol in the cell-free assay described below.

## Epsin TKO cells have defects in cytokinesis

Microscopy of the TKO MEFs revealed them to be abnormally large compared to WT cells (contour in *Figure 1C*). Furthermore, these cells contained multiple clustered nuclei or very large and abnormally shaped nuclei, as shown by both DAPI staining and Cre immunofluorescence (*Figure 1D*, *Figure 1—figure supplement 1B,F*), in contrast to the single nuclei present in the smaller WT cells. This finding could be explained by the reported role of epsin 1 and 2 in mitotic spindle organization (*Liu and Zheng, 2009*), a defect that can result in abnormal chromosome segregation and impaired cytokinesis. Accordingly, analysis of WT and TKO cells during 48-hours of OHT treatment revealed that the increase in either cell size or size/number of nuclei of TKO cells was accompanied by a strong cell number reduction (*Figure 1E*). Additionally, immunostaining of cells synchronized by double-thymidine treatment (*Banfalvi, 2011*) for anti-Aurora B kinase, a marker of midbodies of telophase ([*Banerjee et al., 2014*] inset of *Figure 1D*), showed a striking decrease of cytokinesis profiles in TKO (*Figure 1F*).

As the focus of our study is the role of epsin in endocytosis, we did not explore mechanistic aspects of this cytokinesis defect further. We note, however, that other endocytic proteins, including clathrin itself, have been implicated in the organization of mitotic scaffolds (*Royle, 2013*; *Kaur et al., 2014*).

## Endocytic delay at early/mid-stage clathrin-coated pits

We next analyzed the impact of the complete absence of the three epsins on clathrin-mediated endocytosis. Immunostaining of TKO cells for α-adaptin (a subunit of the clathrin adaptor complex AP-2) and for clathrin-light chain (CLC) showed an increase in the density of clathrin-coated pits relative to controls (*Figure 2A–D*). Additionally, most pits occurred in small clusters rather than as single-puncta (insets in *Figure 2A–D*). Total levels of clathrin and α-adaptin, however, were not changed (*Figure 2—figure supplement 1A*), suggesting that the increase in pits reflects an increase in the pool of assembled clathrin coats (*Figure 2E*). Clathrin-coated pit dynamic was also altered, as live imaging with spinning disk confocal microscopy of cells transfected with μ2-adaptin-GFP (another subunit of the AP-2 complex) revealed that the turnover rate of pits in TKO cells was much lower than in WT cells (*Figure 2F,G*).

In agreement with these morphological changes, an impairment of the internalization of fluorescent transferrin, a cargo of endocytic clathrin-coated pits, was also observed in TKO cells. Following preincubation with Alexa594-transferrin on ice and subsequent incubation at 37°C for 15 min, the bulk of transferrin remained at the cell surface in TKO cells, while remaining transferrin was intracellular in

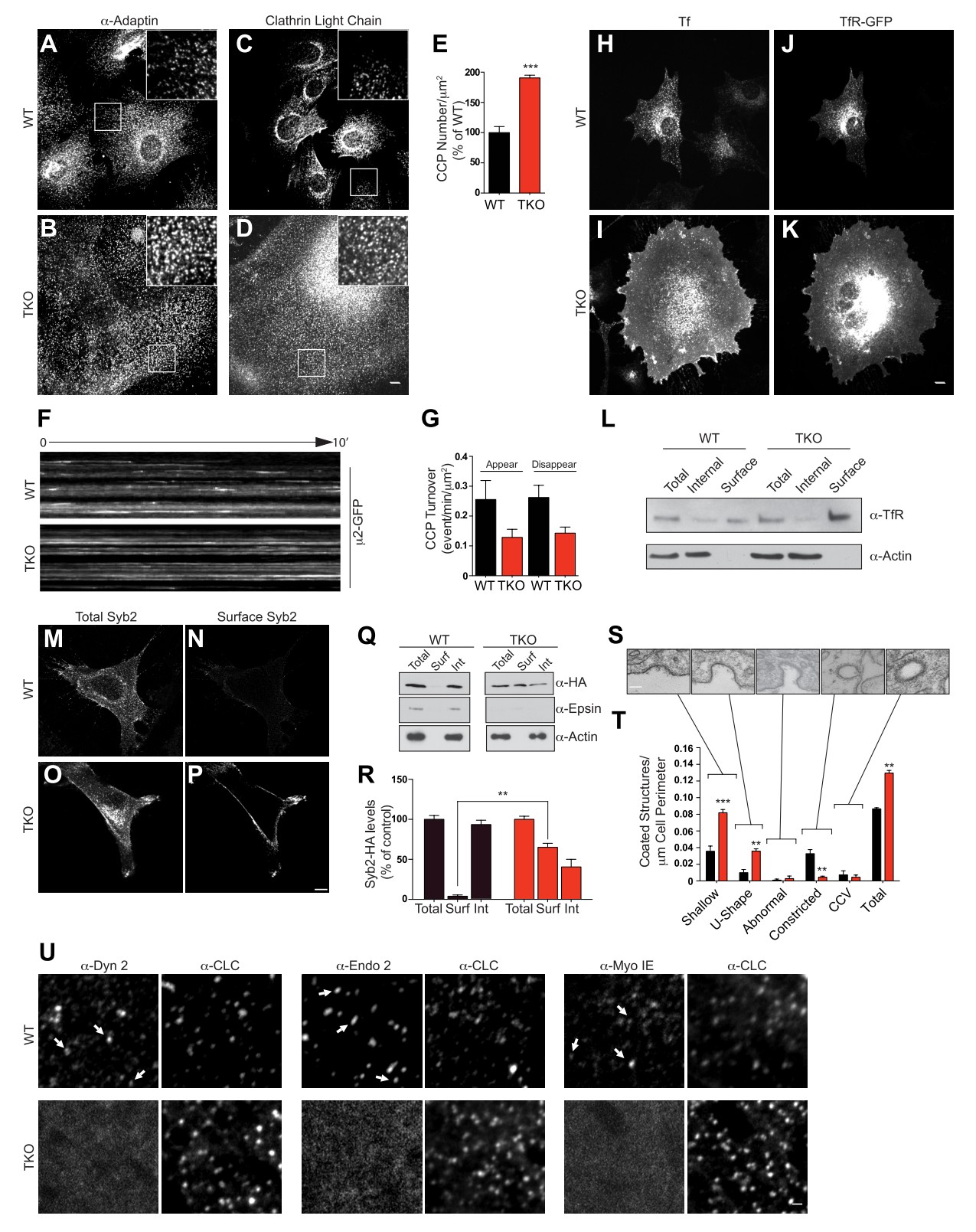

**Figure 2**. Absence of epsin stalls endocytic clathrin-coated pit maturation at an early stage. Confocal microscopy. (**A–D**) Immunofluorescence staining for α-adaptin (**A** and **B**) and clathrin light chain (**C** and **D**) indicates an increase of clathrin-coated pits (CCPs) number in cells that lack all three epsins. In
*Figure 2. Continued on next page*

*Figure 2. Continued*

TKO cells, clathrin-coated pits generally occur in small clusters. Insets show the boxed regions at high magnification. Note the large size of TKO cells relative to control. (**E**) CCP number in WT and TKO cells as assessed by α-adaptin immunofluorescence (***$p < 0.001$, Student's *t* test, n = 10 cells/genotype). (**F**) Kymographs from a time series of WT and TKO cell expressing μ2-adaptin-GFP. Each line represents a single μ2-GFP spot. Note the short length of the lines for WT, reflecting the turnover of the pits and the continuous lines in TKO cells, reflecting an arrest of the pit maturation. (**G**) Clathrin-coated pit turnover (appearance and disappearance events) as analyzed by spinning-disk confocal imaging of μ2-adaptin-GFP fluorescence in WT and TKO cells (n = 5 cells/genotype). (**H and I**) Impaired uptake of pre-bound Alexa594-transferrin (Tf) in TKO cells during a 15-min incubation. In WT, the bulk of Tf was internalized, while in TKO Tf remained at the cell surface. (**J and K**) Transferrin receptor (TfR)-GFP predominantly localizes in intracellular vesicles in WT but at the cell surface in TKO cells. (**L**) A surface biotinylation assay reveals elevated amounts of endogenously expressed TfR at the plasma membrane of TKO cells relative to WT, as assessed by anti-TfR immunoblotting of streptavidin affinity-purified material. (**M–P**) Increased surface localization of stably expressed Syb2-HA in TKO fibroblasts as shown by total (**M–O**) and surface-only (**N–P**) immunofluorescence. (**Q and R**) A surface biotinylation performed as in (**L**) demonstrating an increased fraction of cell surface exposed Syb2-HA in TKO cells relative to WT (**$p < 0.01$, Student's *t* test, n = 4 experiments, Surf: surface, Int: internal). (**S and T**) Representative electron microscopy images of different stage endocytic clathrin-coated intermediates in TKO cells (**S**) and quantification of the corresponding stages (**T**, **$p < 0.01$, ***$p < 0.001$, n = 33 cells/genotype, one-way ANOVA). (**U**) Comparative analysis of the localization of clathrin immunoreactivity (CLC) with the localization of dynamin 2, endophillin 2, and myosin 1E immunoreactivities. In WT cells, these three proteins co-localize with a subset of clathrin-coated pits (examples are indicated by small white arrows), which represent late-stage pits. In TKO cells, where more numerous clathrin-coated pits are observed, the punctate localization of dynamin 2, endophillin 2, and myosin 1E is completely lost. Scale bars: 10 μm for (**A–D**, **H–K**), 20 μm for (**M–P**), 5 μm for (**U**), and 200 nm for (**S**). In **E**, **G**, **R**, and **T** black bars indicate WT and red bars epsin TKO. See also *Figure 2—figure supplement 1*.

The following figure supplement is available for figure 2:

**Figure supplement 1**. Endocytic defect in epsin TKO cells is not due to alteration in endocytic protein levels and it is rescued by epsin1–GFP.

control cells (*Figure 2H,I*). This change correlated with a major shift of overexpressed and endogenous transferrin receptor from a punctate localization in the cytoplasm (WT cells) to a plasma membrane localization (TKO cells), as demonstrated by both immunofluorescence (*Figure 2J,K*) and a cell surface biotinylation assay (*Figure 2L*). Transferrin internalization impairment could be rescued by electroporation of epsin1-GFP cDNA (*Figure 2—figure supplement 1B–D*). Likewise, a major shift of the localization of synaptobrevin 2/VAMP2 (Syb2), a vesicular SNARE that is internalized by clathrin-mediated endocytosis, was observed in TKO MEFs stably expressing Syb2-HA, (this construct harbors the HA epitope at the very short C-terminal non-cytosolic portion of Syb2). Total Syb2 immunoreactivity had a predominant punctate intracellular distribution in control MEFs, and an additional strong plasma membrane localization in TKO (*Figure 2M,O*). The presence of an abundant surface exposed pool of Syb2 selectively in TKO MEFs was confirmed by the surface immunofluorescence for the HA epitope (*Figure 2N,P*) and by a cell surface biotinylation assay (*Figure 2Q,R*).

When cells were examined by electron microscopy to assess the stage at which clathrin-mediated endocytosis is impaired in TKO cells, the major increase was observed for shallow and U-shaped pits (*Figure 2S,T*). This observation may seem in contrast with the reported accumulation of the so-called multiheaded coated-structures at the plasma membrane of cells subjected to RNAi-dependent KD of all the three epsins (*Boucrot et al., 2012*). Such a phenotype was interpreted as a defect in the fission reaction. However, multiheaded-coated structures, which were not clearly detected in our cells (*Figure 2S,T*), could instead reflect clustering of shallow/intermediate stage pits at the pre-fission stage. In agreement with an arrest at early stage, dynamin 2, endophillin 2, and myosin 1E (a myosin implicated in clathrin-mediated endocytosis [*Cheng et al., 2012*]), three proteins whose localization at pits peaks at the time of fission (*Taylor et al., 2011*) were present at the subset of very late pits in WT cells, while in TKO cells they had a diffuse cytosolic distribution (*Figure 2U*).

## Enhanced actin accumulation at epsin-deficient clathrin-coated pits

A stalling of endocytic clathrin-coated pits prior to the deep invagination stage has been observed upon manipulations affecting actin nucleation and dynamics (*Shupliakov et al., 2002*; *Ferguson et al., 2009*), leading us to explore actin localization in TKO cells. Phalloidin staining for F-actin revealed only few of the actin stress fibers typically observed in control cells (*Figure 3A,B*) and a strong accumulation of F-actin foci in TKO cells (*Figure 3B,D*), sometimes in the form of elongated curved structure at the cell cortex (inset in *Figure 3B*). Similar results were observed upon immunostaining for Arp 2/3 (an actin nucleating complex that functions with N-WASP [*Figure 3E*]). A difference in actin organization relative to WT cells was also observed by live total internal reflection

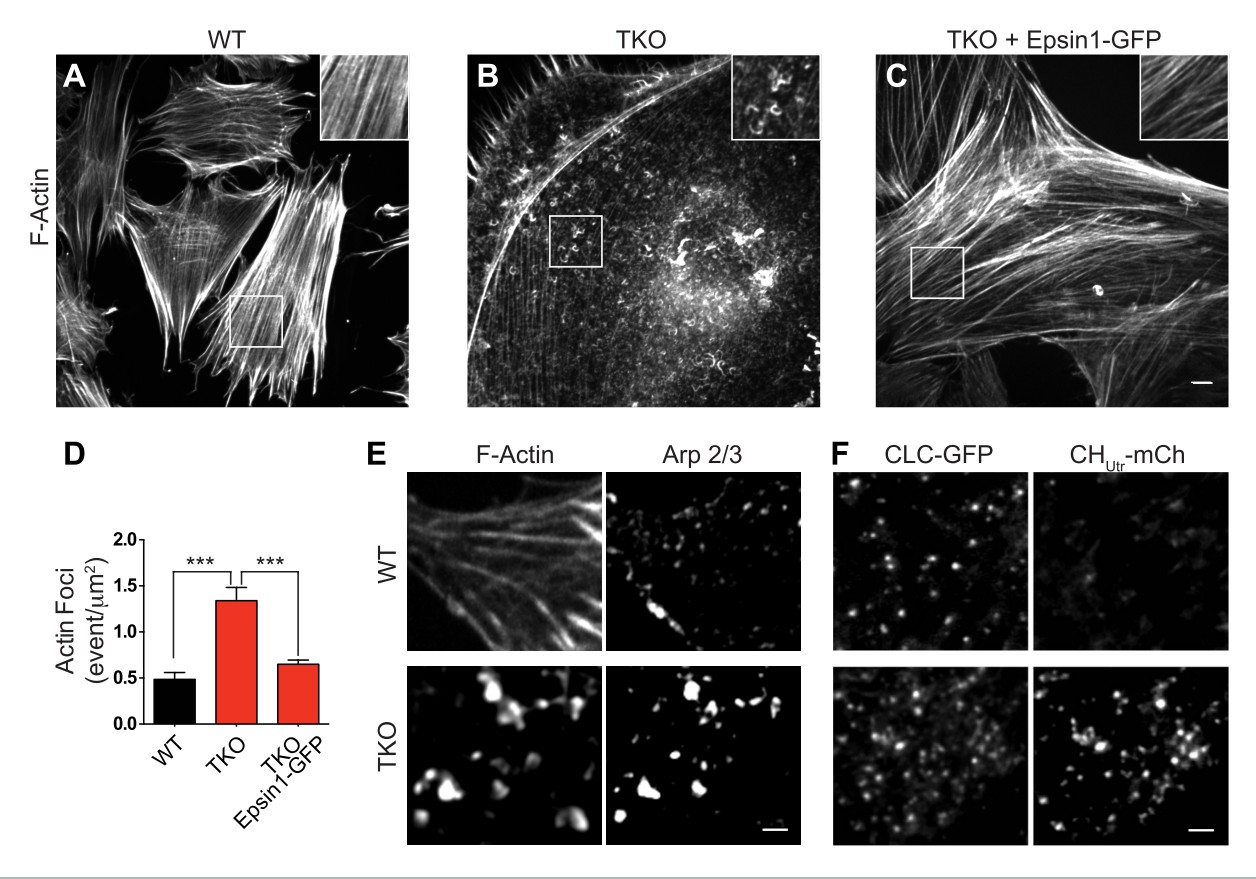

**Figure 3**. Abnormal actin distribution in epsin deficient cells. (A–C) Confocal microscopy images. Phalloidin staining of WT and epsin TKO cells shows a major loss of stress fibers with a corresponding accumulation of elongated F-actin foci (see inset) in TKO cells. These changes were rescued by the expression of epsin1–GFP (C). (D) Quantification of the actin foci shown in (A–C, ***p < 0.001, n = 8 cells/conditions, one-way ANOVA). (E) Phalloidin staining and Arp 2/3 immunoreactivity in a WT and a TKO cell showing the co-localization of Arp 2/3 with the actin foci, as visualized by confocal microscopy. (F) TIRF microscopy of WT and TKO cells expressing clathrin light chain–GFP (CLC–GFP) and the F-actin binding protein utrophin–mCherry (CH$_{Utr}$–mCh). Note the increase in F-actin (CH$_{Utr}$–mCh signal) typically surrounding the clathrin-coated pits, in the cortical region (TIRF plane) of the TKO cell. Virtually all pits are positive for CH$_{Utr}$–mCh. Scale bars: 10 μm for (A–C), 5 μm for (E and F). Data are represented as mean ± SEM.

(TIRF) microscopy of TKO cells, co-expressing the calponin homology domain of utrophin fused to mCherry (CH$_{Utr}$–mCh) and CLC fused to GFP (CLC–GFP), as there was a general increase in the pool of actin associated with clathrin spots (*Figure 3F*). The expression of full-length epsin1–GFP (*Figure 2—figure supplement 1B–D*) rescued actin cytoskeleton changes (*Figure 3C,D*).

## Epsin is required for the recruitment of Hip1R at endocytic pits

Interestingly, while no significant alterations of the levels of several major components of the clathrin-dependent endocytic machinery were observed in TKO cells (*Figure 2—figure supplement 1A*), the level of Hip1R was increased (*Figure 4A*). This observation strengthened the idea that epsin and Hip1R are functionally interconnected, as suggested by studies in yeast and *Dictyostelium* (*Baggett et al., 2003*; *Brady et al., 2010*; *Skruzny et al., 2012*), and prompted us to explore the interplay of epsin and Hip1R.

Pull-down experiments from rat brain homogenate, using ENTH domain of epsin 1 as bait revealed an enrichment of Hip1R in the affinity-purified material in the sample also containing PI(4,5)P$_2$ (*Figure 4B*). Additionally, the typical clathrin-coated pit-like punctate localization (*Engqvist-Goldstein et al., 1999*) of endogenous (immunofluorescence, *Figure 4C*) and exogenous (TIRF microscopy of Hip1R-GFP, *Figure 4F*) Hip1R was replaced in TKO cells by a diffuse signal (*Figure 4D,F*). Expression of epsin1-GFP in TKO cells (*Figure 2—figure supplement 1B–D*) rescued Hip1R localization (*Figure 4E*). We conclude that the localization of HipR at endocytic clathrin-coated pits is heavily dependent on epsin.

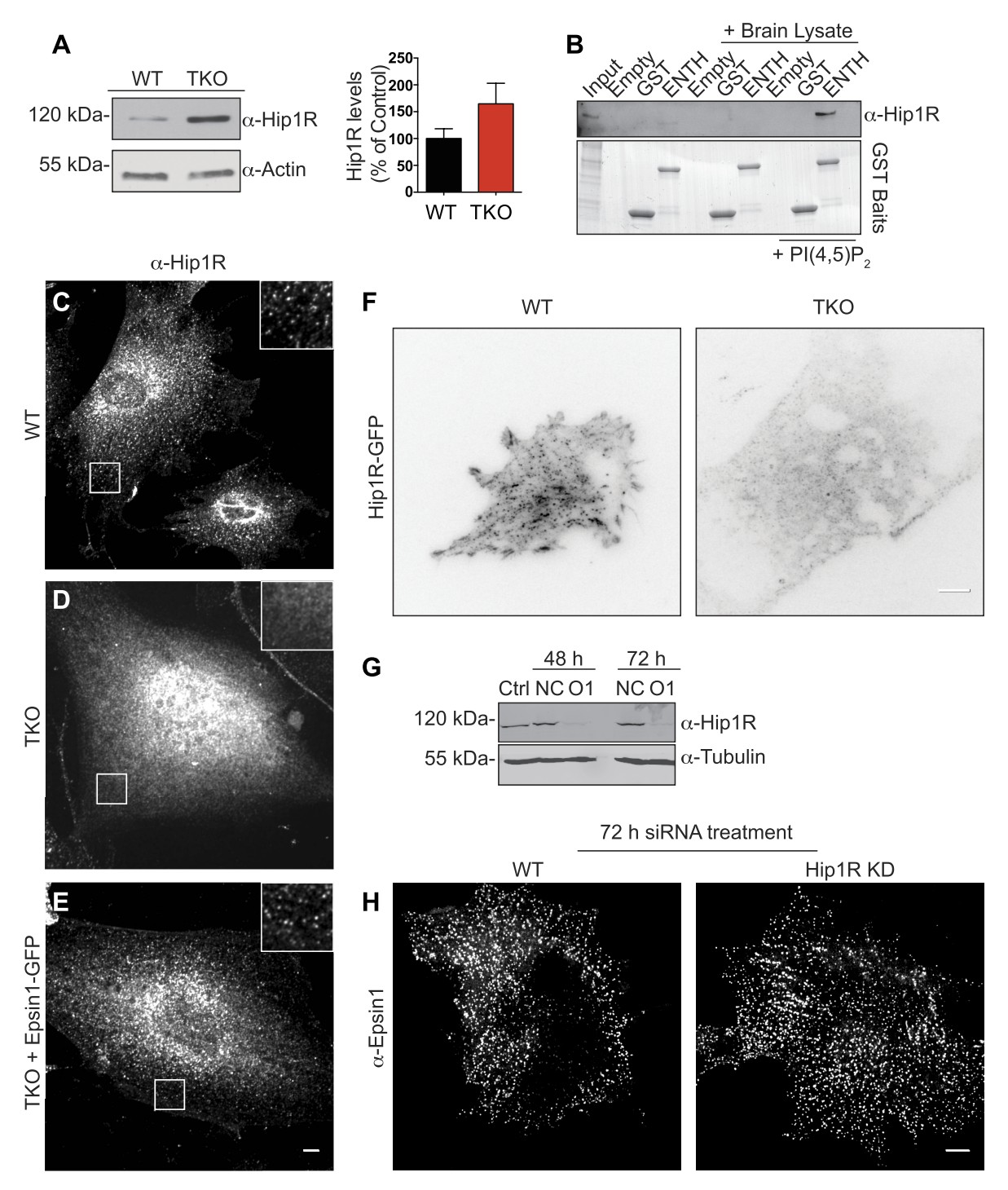

**Figure 4**. Epsin is required for the recruitment of Hip1R at the endocytic clathrin-coated pits. (**A**) Immunoblot analysis of Hip1R (left) shows an increase (as quantified in the right) of its levels in epsin TKO cells. (**B**) Pull down experiments from rat brain homogenate, using ENTH–GST or GST alone as bait, shows the affinity-purification of Hip1R in the presence of diC8-PI(4,5)P$_2$ [PI(4,5)P$_2$, top]. Coomassie blue stained gel of the baits (bottom). (**C–E**) Anti-Hip1R immunofluorescence of representative WT and TKO cells. The clathrin-coated pit pattern of Hip1R in WT (see also high magnification insets) was replaced by a diffuse localization in the TKO cell, but was rescued by expression of epsin1–GFP. (**F**) WT and TKO cells were transfected with Hip1R–GFP and imaged by live TIRF microscopy (fluorescence is shown in black). The punctate accumulation (clathrin-coated pits) of Hip1R at the cortex of WT was lost in the TKO cell. (**G** and **H**) siRNA-mediated knockdown of Hip1R (**G**, Ctrl: non-transfected control; NC: scramble control; O1: Hip1R specific double-stranded siRNA) does not affect epsin localization in HeLa cells as shown by epsin immunofluorescence (**H**). Scale bars represent 10 µm. Data are represented as mean ± SEM.

In contrast, the localization of epsin at pits was independent of Hip1R, as Hip1R KD in HeLa cells by RNAi (*Figure 4G*) did not affect epsin localization (*Figure 4H*).

## Absence of epsin impairs the coupling of clathrin-dependent budding to actin growth in a cell-free assay

To address the role of epsin at the interface of endocytic clathrin coats and actin dynamics, we turned to a cell-free system developed in our laboratory (*Wu et al., 2010*; *Wu and De Camilli, 2012*). The approach consists of incubating glass-attached plasma membrane sheets, produced by cell sonication, with purified brain cytosol in the presence of ATP and the non-hydrolyzable GTP analogue GTPγS. Under these conditions, where endocytic invagination is not compensated by internal pressure, the incubation triggers the formation of endocytic tubules that are capped by clathrin-coated pits and are surrounded by F-BAR proteins of the FBP17 family. Consistent with the role of these proteins in actin nucleation, elongation of the tubules is actin-dependent (*Wu et al., 2010*; *Wu and De Camilli, 2012*).

Plasma membranes obtained by sonication of PTK2 cells expressing PM-anchored GFP exhibit a homogeneous fluorescence when kept in cytosolic buffer (*Wu and De Camilli, 2012*). Upon incubation with WT mouse brain cytosol and nucleotides, growth of the tubular invaginations was revealed in *en face* views by the resulting GFP puncta (*Figure 5—figure supplement 1A*), and 'in side' views by the growth of short columns of GFP fluorescence perpendicular to the plane of the coverslip (*Figure 5A*, left PM-GFP panel). The presence of clathrin at the tips of the tubules was confirmed by immunofluorescence (*Figure 5A*, left clathrin panel). Clathrin signal had an elongated appearance 'in side' views, given the low resolution of confocal microscopy in the Z dimension. Epsin immunoreactivity precisely overlapped with clathrin (*Figure 5A*, left epsin panel) indicating epsin localization throughout the clathrin coat.

Clathrin-capped invaginations also formed on plasma membrane sheets incubated with epsin TKO brain cytosol (*Figure 5A*, right panels and *Figure 5—figure supplement 1B*). However, important differences relative to incubations with control cytosol were observed. In membrane sheets incubated with control brain cytosol, F-actin polymerized in alignment with the invaginations thus generating a well-organized scaffold that contributes to maintain their parallel orientation perpendicular to the plasma membrane (*Figure 5B*, left panels). On the contrary, in membranes incubated with TKO cytosol, the F-actin network around the invaginations was exaggerated and disorganized (*Figure 5B*, right panels), leading to clumping of the tips of the invaginations in a meshwork of actin. As a result of this clumping, invaginations often had an oblique orientation (*Figure 5A,B*, right panels and *Figure 5—figure supplement 1B*). Such disorganization was confirmed by electron microscopic observations of sections cut parallel to the substrate. Regularly spaced cross-sectioned tubules interspersed within an actin meshwork were observed with WT cytosol, but numerous obliquely cut tubules embedded in a dense actin matrix were visible with TKO cytosol (*Figure 5—figure supplement 1C,D*). Another striking modification related to an abnormal actin cytoskeleton was the mislocalization of myosin 1E. As shown by immunofluorescence, this protein was localized at the bottom of the tubular invaginations in sheets incubated with WT cytosol, but at the tips in sheets incubated with epsin TKO brain cytosol (*Figure 5C* and higher magnifications).

## Epsin is required for the recruitment of Hip1R to endocytic invaginations in the cell-free assay

The differences in the cell-free assay using WT vs TKO cytosol were consistent with abnormal actin nucleation observed at clathrin-coated pits of intact TKO cells. Furthermore, in agreement with these observations, Hip1R immunoreactivity was present at the tips of the invaginations in sheet preparations incubated with WT cytosol, but was largely absent from the tubules incubated with TKO cytosol (*Figure 5D*). This was in spite of the higher levels of Hip1R in the TKO brain cytosol (*Figure 5—figure supplement 1E*) in agreement with the higher levels of Hip1R in TKO fibroblasts (*Figure 4A*). Thus, also in this system, Hip1R recruitment requires epsin.

We also prepared cytosol from brains of mice lacking Hip1R and its close homologue Hip1 (Hip1/Hip1R double KO (DKO) mice [*Bradley et al., 2007*]). These mice are typically dwarfed, afflicted with severe spinal defects, and die in early adulthood (*Bradley et al., 2007*). When membrane sheets were incubated with Hip1/Hip1R DKO brain cytosol, differences from WT were observed that were qualitatively similar to those obtained with the epsin TKO cytosol, but quantitatively milder (*Figure 5E,F*). Tubular invaginations were longer and some exaggerated and disorganized actin was also observed

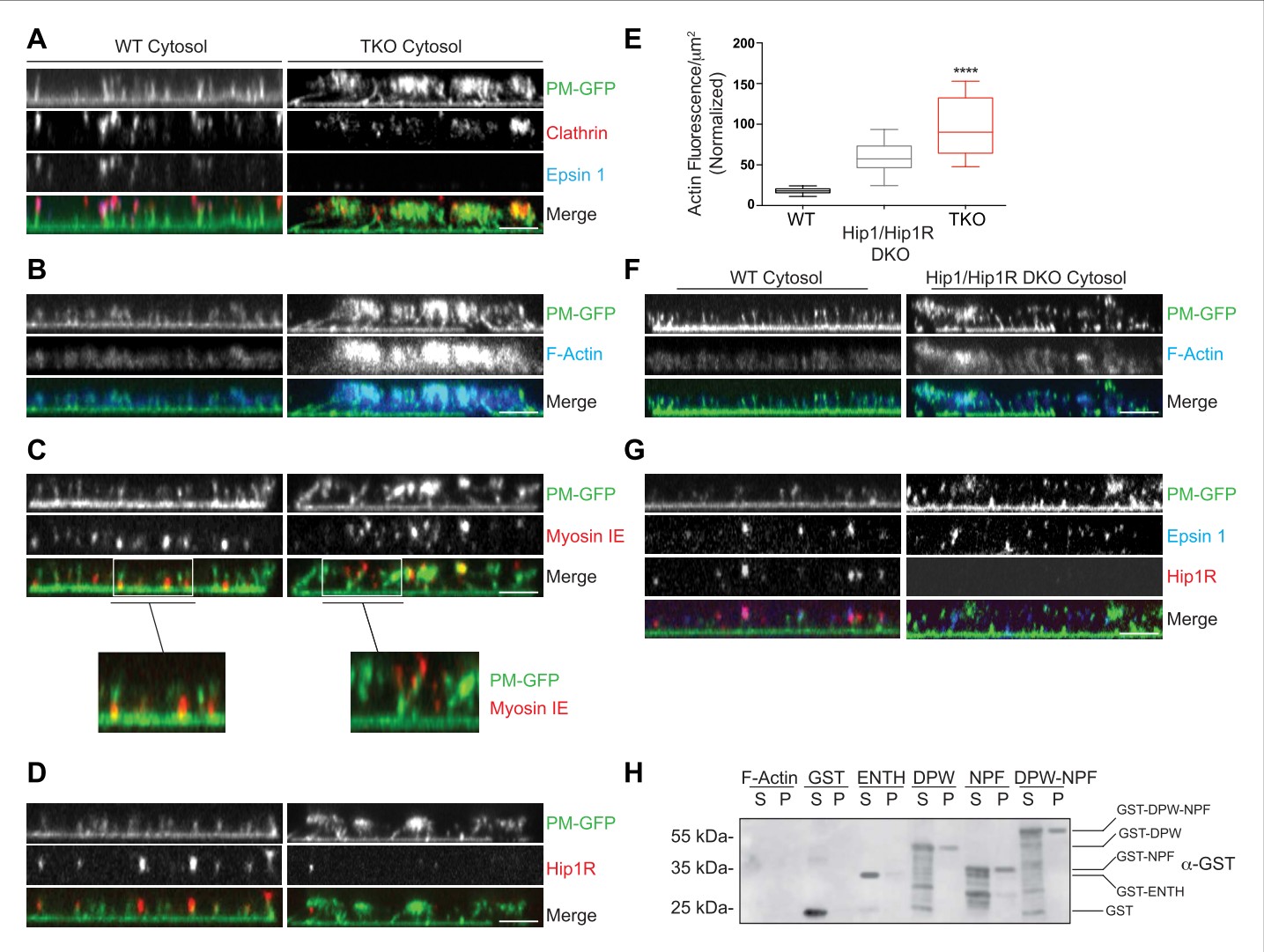

**Figure 5**. Coupling of clathrin-dependent budding to actin dynamics in a cell-free assay is perturbed by the absence of epsin. Plasma membrane sheets of PTK2 cells expressing PM-anchored GFP (PM–GFP) were incubated in the presence of WT, epsin TKO or Hip1/Hip1R double KO (DKO) brain cytosol, and nucleotides as described in *Wu et al. (2010)*, fixed, immunostained, and observed by confocal microscopy. Views orthogonal to the substrate are shown. (**A**) Immunofluorescence staining for clathrin light chain and epsin 1. The GFP-positive columns represent narrow tubular invaginations of the plasma membrane capped by clathrin-coated pits, as revealed by the presence of clathrin and epsin immunoreactivity (*Wu et al., 2010*). In the control preparation (left) tubules are straight and perpendicular to the substrate, while in the preparation incubated with TKO cytosol (right, note the absence of epsin immunoreactivity) they have a more disordered orientation. (**B**) Phalloidin staining of the membrane sheets revealing a well-organized actin scaffold around the tubules after incubation with WT cytosol (left) and an exaggerated and disorganized F-actin network in sheets incubated with TKO cytosol (right). (**C**) Myosin 1E immunoreactivity localizes at the bottom of tubular invaginations in sheets incubated with WT cytosol (left), and at the tip of the invaginations after incubation with TKO cytosol (right). Details of the sheets are shown at high magnification below the main panels. (**D**) Hip1R immunoreactivity is present at the tip of tubular invaginations in the control samples (left), but is absent in sheets incubated with epsin TKO cytosol (right). (**E**) Quantification of F-actin polymerization upon incubation with the cytosol of the three tested genotypes (WT, epsin TKO, and Hip1/Hip1R DKO). The average phalloidin fluorescence per unit area of membrane sheets was calculated (n = 10 sheets/conditions, ****p < 0.0001, one-way ANOVA). (**F**) Phalloidin staining of sheets incubated with Hip1/Hip1R double KO (right) cytosol reveals an exaggerated and disorganized F-actin network relative to WT (left), but not as prominent as that observed in preparations incubated with epsin TKO cytosol. (**G**) Immunofluorescence staining for epsin 1 and Hip1R shows that the presence of epsin 1 at the tips of the invagination is not strongly modified by the absence of Hip1 and Hip1R. (**H**) The disordered C-terminal tail of epsin binds F-actin. GST fused epsin 1 fragments (ENTH domain, DPW, NPF, and DPW-NPF containing regions) were incubated (5 µM final concentration) with previously polymerized F-actin (15 µM) and then subjected to ultracentrifugation followed by SDS-PAGE and anti-GST immunoblotting of the supernatant (S) and pellet (P) materials. Scale bars: 5 µm. Data are represented as mean ± SEM. See also *Figure 5—figure supplement 1*.

The following figure supplement is available for figure 5:

**Figure supplement 1**. Nestin Cre epsin TKO brain cytosol has increased Hip1R level and disrupted membrane tubulation in a cell-free assay.

(*Figure 5E,F*). Importantly, localization of epsin 1 at the tips of the invaginations was not affected by the absence of Hip1/Hip1R in the cytosol (*Figure 5G*), once again as in the case of intact cells.

## Direct binding of epsin to actin

Altogether, the results discussed above support a role for epsin as a critical factor required for the temporal and spatial coordination between clathrin-mediated endocytosis and actin dynamics. We thus explored whether epsin interacts directly with actin by incubating recombinant epsin 1 fragments (*Figure 5—figure supplement 1F*) with purified F-actin followed by co-sedimentation. This analysis revealed that both the DPW and NPF motif containing regions of the protein (*Chen et al., 1998*), but not the ENTH domain, bind actin (*Figure 5H*). Thus, there appear to be at least two actin-binding sites in epsin. The site in the NPF motif containing region likely corresponds to the actin cytoskeleton-binding (ACB) site, previously identified in yeast epsin (Ent1, *Skruzny et al., 2012*).

## A low affinity binding between epsin and synaptobrevin 2/VAMP2

Collectively, previous findings from the literature and the findings described above are consistent with roles for epsin from early to late stage clathrin-coated pits, and more generally for an essential role of this protein in clathrin-mediated endocytosis in mammalian fibroblasts. It was therefore of interest to determine whether epsin, like several clathrin adaptors with ENTH/ANTH domains (*Hirst et al., 2004*; *Miller et al., 2007, 2011*; *Koo et al., 2011*), binds SNAREs. SNARE binding by ENTH/ANTH family proteins helps coordinate bud formation with the incorporation of an appropriate SNARE in the nascent vesicle to direct its fate after fission. Such binding in the case of epsin would likely be of very low affinity, as it was not detected in previous investigations. However, even a low affinity binding could be of physiological significance in the context of other synergistic interaction between epsin, the membrane bilayer and other coat proteins.

Pull-downs from mouse brain lysate using GST–ENTH domain of epsin 1 as bait, followed by analysis of the affinity purified material by mass spectroscopy, identified Syb2 as one of the top hits (*Figure 6—figure supplement 1A*). A direct interaction was next explored by assessing binding to the ENTH domain of epsin 1 (with an His tag at the C-terminus) to increasing concentrations of the cytosolic portion of Syb2 fused to GST (*Figure 6A*). The very low affinity of this interaction ($K_D$ 470 ± 30 μM) was enhanced by the presence in the incubation medium of $IP_6$ ($K_D$ 80 ± 7.9 μM, *Figure 6B,D* and *Figure 6—figure supplement 1B*), which induces the folding of helix zero of the ENTH domain (*Itoh et al., 2001*; *Ford et al., 2002*). In contrast, the deletion of helix zero from the ENTH construct abolished Syb2 binding in either the presence or the absence of $IP_6$ (*Figure 6C*). Further analysis of the interaction of Syb2 with the ENTH domain using Syb2 fragments (*Figure 6E*) identified the SNARE motif of Syb2 (*Figure 6F,G*), and more specifically the N-terminal portion of the motif (residues 29–60, *Figure 6H,I*), as the region responsible for binding. Interestingly, previously reported interactions of vesicular SNAREs with ENTH/ANTH proteins also involve their SNARE motifs (*Koo et al., 2011*; *Miller et al., 2011*), pointing to an interaction mutually exclusive with SNARE complex formation, as expected for an 'endocytic' interaction.

The interaction of epsin with Syb2 also occurs in intact cells, as shown by anti-HA co-immunoprecipitation experiments from HeLa cells expressing FLAG-tagged epsin 1 and either HA-tagged full-length Syb2 (*Figure 6—figure supplement 1C*), or deletion constructs of Syb2 (*Figure 6—figure supplement 1D*). Only constructs including the N-terminal portion of the SNARE motif of Syb2 co-precipitate epsin 1 (*Figure 6—figure supplement 1E*).

The low affinity interaction of epsin with an exocytic SNARE is evolutionarily conserved as yeast Snc1 (a Syb2 homologue) bound the ENTH domain of Ent1 in a recombinant pull-down assay (*Figure 6J*).

## Discussion

Previous reports have addressed the role of epsin in endocytosis. However, studies in different organisms and in cell-free systems had emphasized the different aspects of its function, so that a complete picture of its physiological role(s) has remained elusive. Elucidation of epsin function in mammalian cells had been complicated by the existence of the three *Epn* genes with overlapping functions. To overcome this problem, we have generated cells that lack all the three epsins by a conditional KO approach involving the OHT-dependent disruption of the *Epn1* gene in epsin 2 and 3 double KO mice.

A main conclusion of our study is the occurrence of a close coupling between the function of epsin and the dynamics of the actin cytoskeleton. While such coupling had been suggested by studies in unicellular organisms, studies of epsin in cells of metazoan had emphasized its role as a bilayer

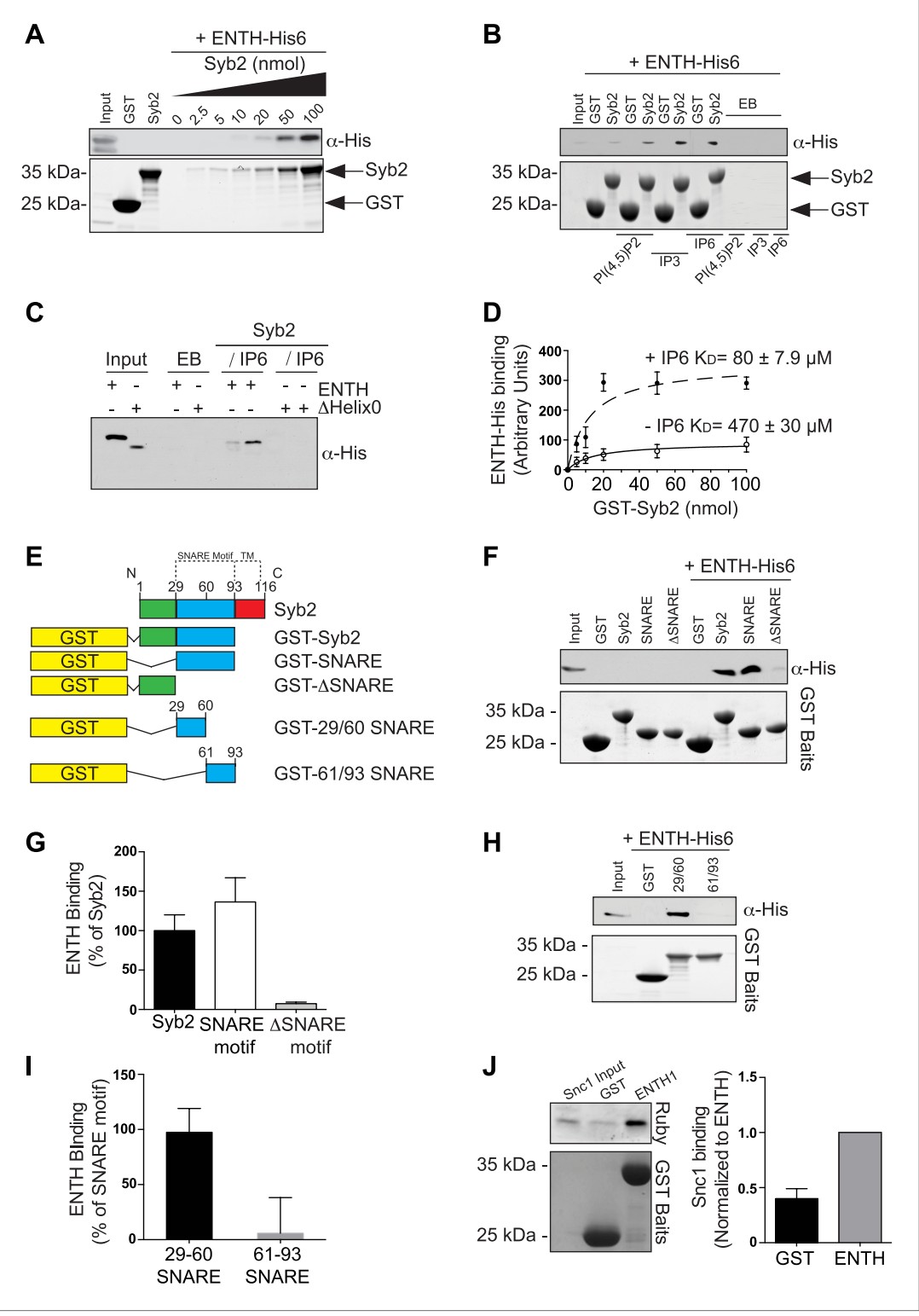

**Figure 6**. Epsin directly binds synaptobrevin 2/VAMP2. (**A**) Interaction of purified epsin ENTH–His6 (800 ng) with increasing amounts of GST-cytosolic portion of synaptobrevin2 (Syb2) as revealed by anti-His immunoblotting of bound material in a GST pull-down (top). Coomassie blue stained gel of the baits (bottom). (**B**) The interaction of epsin ENTH–His6 with equal amounts of GST–Syb2 (Syb2) is enhanced by the presence of soluble diC8-PI(4,5)$P_2$ (PI(4,5)$P_2$), IP$_3$ or IP$_6$ (final concentration 50 µM). Top: anti-His immunoblotting. Bottom: Coomassie blue stained

*Figure 6. Continued on next page*

*Figure 6. Continued*

gel of the baits. EB: empty beads. (**C**) The interaction of epsin ENTH–His6 with equal amounts of GST–Syb2 is lost in a construct missing the N-terminal helix zero (ΔHelix0) irrespective of the presence of IP$_6$ (final concentration 50 μM). EB: empty beads. (**D**) Quantitative analysis of the binding of ENTH–His6 to increasing amounts of GST–Syb2 in the presence or absence of IP$_6$, as revealed by densitometry of anti-His immunoreactivity in western blots of bound material. K$_D$s are also indicated. (**E**) Schematic representation of constructs used for (**F–I**). SNARE motif and transmembrane region (TM) are denoted by dotted lines. (**F** and **G**) Pull-down of ENTH–His6 by GST fusions of different cytosolic fragments of Syb2. Anti-His immunoblotting (**F**) shows binding only to SNARE motif containing fragments (top). Coomassie blue stained gel of the baits (Bottom). Quantitative analysis of the results is shown in (**G**). (**H** and **I**) ENTH–His6 domain binding to SNARE motif fragments (**H**) and corresponding quantification (**I**). Anti-His immunoblotting detects ENTH interaction with the N-terminal portion of the SNARE motif. (**J**) Purified yeast His-Snc1 binds a GST fusion of the ENTH domain of Ent1 (yeast epsin 1) in a pull down assay (top). Coomassie blue stained gel of the baits (Bottom). Corresponding quantification is shown. Data are represented as mean ± SEM. See also *Figure 6—figure supplement 1*.

The following figure supplement is available for figure 6:

**Figure supplement 1**. Epsin binding to Syb2: mass spectroscopy analysis and interaction in intact cells.

---

deforming protein (*Ford et al., 2002*; *Boucrot et al., 2012*) and as a cargo-specific adaptor (*Polo et al., 2002*; *Overstreet et al., 2003*; *Chen and De Camilli, 2005*; *Meloty-Kapella et al., 2012*). A link of epsin to actin function at endocytic clathrin-coated pits, and in particular a role as a factor that ensures a 'controlled' actin growth at sites of endocytosis, is supported by several observations. First, Hip1R, which had so far been considered the major link of endocytic clathrin coats to actin in mammalian cells (*Engqvist-Goldstein et al., 2001*; *Le Clainche et al., 2007*), is not recruited to endocytic clathrin-coated pits in the absence of epsin. This is consistent with the interaction and functional partnership of epsin with Sla2/Hip1R, previously demonstrated in yeast and *Dictyostelium*, although in yeast it is Sla2 that recruits epsin to clathrin-coated pits (*Skruzny et al., 2012*). Second, the lack of epsin results in a disruption of the normal actin cytoskeleton with a major loss of stress fibers and an accumulation of actin foci at the cell surface, typically in proximity of arrested endocytic clathrin-coated pits. Third, epsin binds actin directly via its disordered tail region. Fourth, a cell-free assay of endocytosis involving plasma membrane sheets and brain cytosol revealed that relative to WT, epsin-deficient cytosol results in an exaggerated and abnormal growth of the actin cytoskeleton. These findings suggest a role for epsin in orchestrating a coupling between actin and endocytic clathrin coats that (1) limits excessive actin growth and (2) helps mediate an effect of actin polymerization on bud invagination. A functional partnership between epsin and Hip1R explains why the KD of Hip1R results in a similar enhanced and abnormal actin nucleation at endocytic sites, with the formation of large actin bundles projecting away from these sites (*Engqvist-Goldstein et al., 2004*).

The excess of actin nucleation in the absence of epsin is of special interest as in yeast the ENTH domains of Ent1 and 2 bind two Cdc42–GAPs (GTPase activating proteins), Rga1 and Rga2, and this interaction is essential for function (*Aguilar et al., 2006*). Interestingly RLIP76/RalBP1, a protein with Cdc42 GAP activity, interacts with the ENTH domain of epsin (*Rossé et al., 2003*), although there is no clear sequence homology between RLIP76/RalBP1 and Rga proteins. In view of the modulatory role of Cdc42 in the nucleation of N-WASP-Arp 2/3-dependent actin nucleation at endocytic clathrin-coated pits, the recruitment of a Cdc42–GAP by epsin could help explain why actin nucleation is enhanced in epsin null cells.

A very strong link between epsin and actin is consistent with evidence for a general and evolutionarily conserved role of actin in clathrin-mediated endocytosis, which is more prominent under conditions of high plasma membrane tensions (*Merrifield et al., 2005*; *Kaksonen et al., 2006*; *Ferguson et al., 2009*; *Boulant et al., 2011*; *Taylor et al., 2012*). The force of actin polymerization may help drive the deep invagination of the pits and also assist in dynamin-dependent fission (*Itoh et al., 2005*; *Roux et al., 2006*) by developing a force that drives the deeply invaginated bud away from the plasma membrane. The occurrence of this force is well demonstrated by the dramatic actin-dependent elongation of the clathrin-coated pit necks in living cells and in cell-free systems (*Ferguson et al., 2009*; *Wu et al., 2010*), when the fission reaction is impaired by the lack of dynamin (living cells) or block of its GTPase activity (cell-free system), respectively. Clearly, however, epsin does not simply function as

a negative regulator of actin polymerization, but it is required for proper spatial organization of the actin-based cytoskeleton, as exemplified by the abnormal localization of myosin 1E at sites of endocytic invaginations on plasma membrane sheets incubated with epsin TKO cytosol.

We note that partial deficiency of epsin function has a major impact on intercellular signaling mediated by Notch (*Overstreet et al., 2003*; *Chen et al., 2009*). It was proposed that Notch activation requires a pulling force applied to Notch by the endocytosis of its ligand in the neighboring cell and that epsin may be essential for the generation of such force (*Meloty-Kapella et al., 2012*; *Musse et al., 2012*; *Shergill et al., 2012*). Based on our findings, such a requirement for epsin may be linked to its role in the control of actin dynamics at clathrin-coated pits. Reported roles of epsin in clathrin-independent endocytosis may also be linked to its function in actin regulation (*Sigismund et al., 2005*).

Another important finding of our study is a weak interaction of epsin's ENTH domain with the SNARE synaptobrevin 2/VAMP2. As intracellular vesicles need SNAREs to fuse with their target membranes, a coupling must exist between vesicle budding and incorporation of the proper SNARE(s) in the bud. Accordingly, other coat adaptors with ANTH or ENTH modules, such as AP180/CALM and epsinR were shown to bind SNAREs (*Hirst et al., 2004*; *Miller et al., 2007, 2011*; *Koo et al., 2011*). We hypothesize that the very low affinity of the binding of synaptobrevin 2 to epsin, which explains why such an interaction had not been described so far, may nevertheless be physiologically relevant when compounded by other interaction of epsin with the bilayer, other coat components, and ubiquitinated cargo proteins.

Recently, it was proposed that a main role of epsin in endocytic clathrin-coated pit dynamics is to help mediate membrane fission via the membrane remodeling properties of the amphipathic helix zero of its ENTH domain (*Boucrot et al., 2012*). While this action seems plausible, our study strongly suggests that epsin becomes critically important at the earlier stages of clathrin coat maturation, as we have observed an accumulation of shallow and U-shaped endocytic clathrin-coated pits in epsin TKO cells. The diffuse, rather than punctate distribution of proteins that assemble at the necks of deeply invaginated endocytic clathrin-coated pits, such as dynamin 2, endophilin 2, and myosin 1E further supports a stalling of pits at an early stage in epsin TKO cells. In contrast to the observations made in epsin triple KD cells, we have not observed clearly multiheaded pits, although the dome of some pits was irregular. Multiheaded pits observed in epsin triple KD cells had been interpreted as reflecting a defect in fission. However, they do not have the constricted neck of multiheaded pits observed in dynamin mutant cells (*Ferguson et al., 2007, 2009*) and at least some of them could represent a clustering of shallow or U-shaped pits. We note that the multiple endocytic functions of epsin are consistent with its localization throughout the coat (*Hawryluk et al., 2006*; *Sochacki et al., 2014*), rather than a selective localization at the bud neck to mediate fission.

Finally, our work also demonstrates a dramatic impact of the lack of epsin on cell division, extending previous observations made in epsin 1 and 2 double KD cells, where abnormal spindles were detected (*Liu and Zheng, 2009*). Surprisingly, we did not find a major effect on cell division in cells derived from epsin 1 and 2 double KO embryos possibly because of long-term adaptation to decreased levels of epsin. For example, we have detected an upregulation of epsin 3 in epsin 1 and 2 double KO cells (our unpublished observations). Interestingly, epsin undergoes phosphorylation and mono- or oligo-ubiquitination (*Stukenberg et al., 1997*; *Chen et al., 1999, 2003*; *Polo et al., 2002*) in mitosis and these covalent modifications impair its binding to clathrin and AP-2, suggesting a switch of its function in the mitotic cytosol (*Chen et al., 1999*). A role of epsin in the dynamics of the cytoskeleton may underlie its role in mitosis, for example by affecting its anchoring to the cell cortex. Several other endocytic proteins, including clathrin, are also implicated in mitosis, and, in particular, in cytokinesis (*Royle, 2013*).

In conclusion, our results reveal new aspects of the collective function of the three epsin genes and point to epsin as an important coordinator of the dynamics of endocytic clathrin coats from early to late stages. Our findings, along with previous studies suggest the models depicted in *Figure 7*. Epsin is an early component of endocytic clathrin-coated pits, which interacts directly with PI(4,5)P$_2$, AP-2, clathrin, and other early clathrin-coated pit components such as intersectin and Eps15. The partial bilayer penetration of its ENTH domain makes it optimally suited to function at a site where membrane buckling occurs (as a generator, sensor or stabilizer of curvature), while its low affinity binding to synaptobrevin 2/VAMP2 may collaborate with AP180/CALM to ensure the coupling of bud nucleation to a membrane patch containing SNAREs. It may additionally participate in the recruitment to the pit of ubiquitinated cargo proteins via its UIMs, a function that does not apply to *Dictyostelium* epsin,

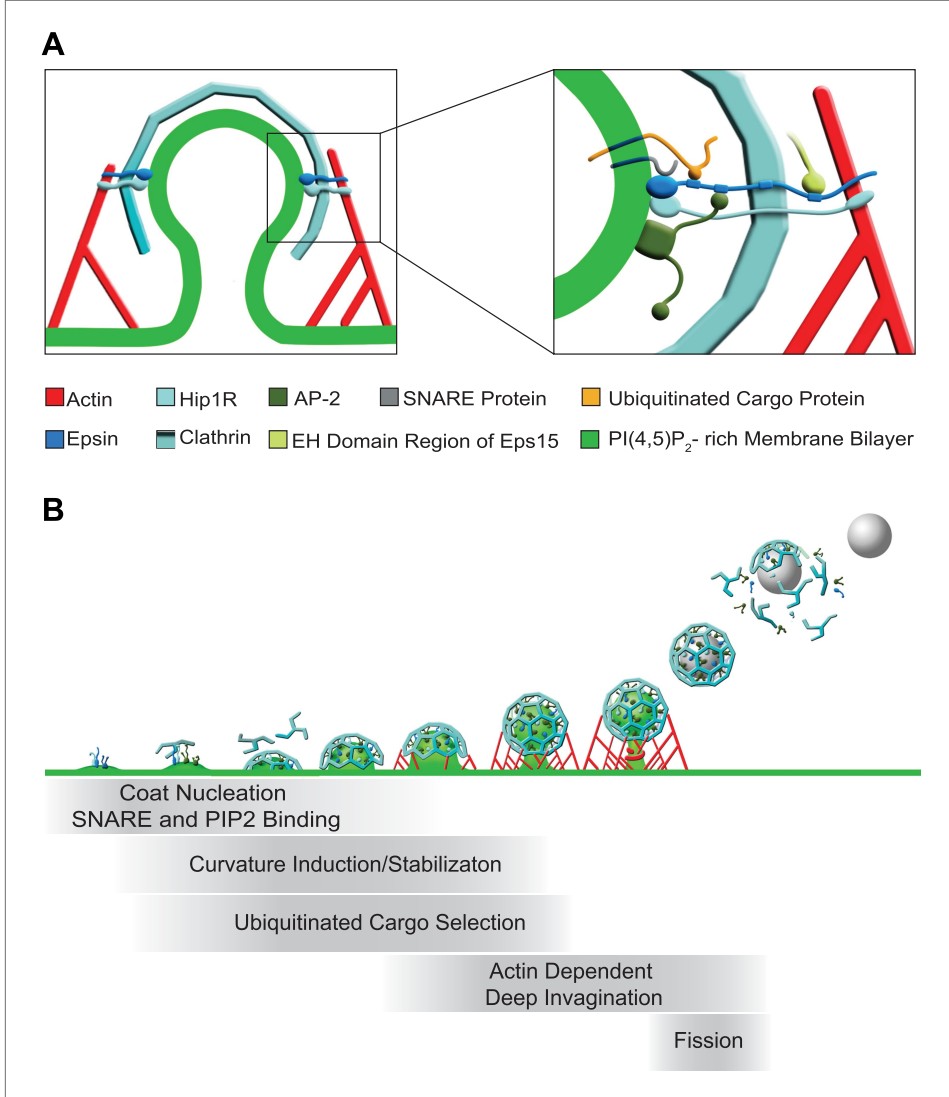

**Figure 7**. Functions of epsin in clathrin-mediated endocytosis. (**A**) Schematic representation of the role of epsin in the coupling of the endocytic clathrin coat to actin in cooperation with Hip1R (left). This coupling helps invaginate the pit. Only epsin localized at the equator of the bud is shown to emphasize its actin-related function in pit invagination, but epsin is not restricted to the equator. Higher magnification representation of the interactions of epsin (right): the N-terminal ENTH domain binds (1) the PI(4,5)$P_2$-rich membrane bilayer of the plasma membrane (and partially penetrates it), (2) the ANTH domain of Hip1R, and (3) synaptobrevin2/VAMP2; its tail binds (1) ubiquitinated cargo proteins (via UIMs), (2) AP-2 and clathrin heavy chain (via DPF/DPW motifs and 'clathrin boxes', respectively), and (3) the EH domain region of Eps15 (via NPF motifs). Epsin's tail also binds F-actin and co-operates with the THATCH domain of Hip1R in the coupling of the actin cytoskeleton to the clathrin-coated pit. In both fields Hip1R is depicted as a monomer but functions as a dimer, or heterodimer with Hip1. (**B**) Schematic representation of the multiple functions of epsin in clathrin-mediated endocytosis. Epsin participates in the early stages of the reaction as a coat nucleator, curvature inducer/sensor and also helps coupling bud formation to SNARE incorporation. As the coat matures, ubiquitinated cargo is recruited. Additionally, the link between epsin/Hip1R and actin is required for the deep invagination of the pit, and this is the process at which the action of epsin becomes essential. Subsequently, the force produced by actin and the bilayer destabilizing properties of epsin's ENTH domain may co-operate with dynamin in fission.

which expresses a single epsin family protein lacking UIMs. As the bud expands, epsin along with Hip1R provide a link between actin nucleation and coat maturation, which is required for deep invagination. Finally, the membrane insertion properties of its helix zero may assist dynamin in the fission

reaction. As we show here, however, its most critical function is at the transition between shallow/U-shaped and deep invaginated pits.

## Materials and methods

### Antibodies and reagents

Antibodies were obtained from the following commercial sources: mouse anti-tubulin, rabbit anti-FLAG, and mouse anti-Actin (Sigma-Aldrich, St. Louis, MO, USA); mouse anti-transferrin receptor and rabbit anti-GFP (Life Technologies, Carlsbad, CA, USA); goat anti-epsin 1 and mouse anti-GST (Santa Cruz Biotechnology, Santa Cruz, CA, USA); rabbit anti-clathrin light chain, mouse anti-Arp 2/3, rabbit anti-Hip1R, and mouse anti-Hip1 (EMD-Millipore, Billerica, MA, USA); rat HRP-conjugated anti-HA (Roche, Mannheim, Germany); rabbit anti-His tag (GenScript, Piscataway, NJ, USA); mouse anti-Aurora B, rabbit anti-caveolin-1, and mouse anti-adaptin µ2 subunit (BD Transduction Laboratories, San Jose, CA, USA). The following antibodies were kind gifts: mouse anti-epsin 1 and mouse anti-epsin 3 (Pier-Paolo Di Fiore, IFOM, Milan, Italy); rabbit anti-Hip1R (David Drubin, University of California, Berkeley, CA, USA); rabbit anti-Myosin 1E (Mark Mooseker, Yale University, New Haven, CT, USA) and rabbit anti-synaptophysin (Paul Greengard, The Rockefeller University, New York, NY, USA). Rabbit anti-epsin 2, mouse anti-clathrin heavy chain, mouse anti-adaptin α subunit, rabbit anti-dynamin 2 and rabbit anti-endophillin 2 were generated in our lab. Alexa594-phalloidin, Alexa594-mouse anti-HA and Alexa488, Alexa594, and Alexa647 conjugated secondary antibodies were from Life Technologies; Alexa405-phalloidin was from AAT Bioquest (Sunnyvale, CA, USA) while DAPI was from Sigma-Aldrich.

### Plasmids

C-terminal GFP-tagged epsin 1 was obtained by PCR amplification of the epsin 1 coding sequence from brain cDNA library (Clontech Laboratory Inc., Mountain View, CA, USA) followed by ligation into the pEGFP–N3 plasmid (Clontech Laboratory Inc.).

For the generation of the GST tagged ENTH, the rat epsin 1 ENTH coding sequence was PCR amplified and ligated in the pGEX4T-1 vector (GE Healthcare, Pittsburgh, PA, USA).

The GST-tagged epsin fragments (ENTH, DPW, NPF) were generated by the ligation of the corresponding PCR products to pGEX4T-1 (GE Healthcare), while for the GST-tagged DPW–NPF tandem fragment the ligation was with pGEX6 (GE Healthcare).

The 6xHis-tagged ENTH of rat epsin 1 was produced by cloning ENTH coding sequence in pET21a(+) (EMD-Millipore), and the resulting coding plasmids was mutagenized by site directed mutagenesis (QuikChange II XL, Agilent Technology, Santa Clara, CA, USA) to remove the helix zero coding sequence in order to obtain ΔHelix0-ENTH His6.

GST-tagged Syb2 was created by PCR amplification of the cytosolic portion of human Syb2, the corresponding SNARE motif or the N-terminal extra SNARE region. PCR products were ligated in the pGEX4T-1 vector (GE Healthcare).

FLAG-tagged full-length epsin 1 was generated by cutting epsin 1 coding sequence from the above reported epsin1-GFP followed by ligation in the pFLAG-CMV4 (gift from Angus Nairn, Yale University, New Haven, CT, USA).

For the generation of a cell line stably expressing Syb2-HA, human Syb2 coding sequence in fusion with HA was cut from pCI neo Syb2-HA and cloned into pBABE puro. Syb2-HA fragments (Δ1–31, Δ1–50, Δ1–70) were generated by site directed mutagenesis (QuikChange II XL, Agilent Technology) of pCI neo Syb2-HA. For study of yeast Ent1 ENTH domain–Snc1 interaction, the following plasmids were used: GST control (pBW1546 pGEX-5X-1), WT ENTH1 (pBW1800 pGEX-5X-1:ENTH1), and Snc1 cytosolic tail (pBW1916 pET28a:Scn1 [aa1-93]).

pGST 29/60 and pGST 61/93 Syb2 SNARE were a gift from Karin Reinisch (Yale University, New Haven, CT, USA); Hip1R–GFP was a gift from David Drubin (University of California, Berkeley, CA, USA); and clathrin light chain–GFP was a gift from James Keen (Thomas Jefferson University, Philadelphia, PA, USA). Adaptin µ2 subunit-GFP, mCherry-tagged utrophin, and GFP-tagged human transferrin receptor were previously described (*Merrifield et al., 2005*; *Zoncu et al., 2007*; *Ferguson et al., 2009*).

### Generation of epsin conditional triple knock out mice

Conditional *Epn1* KO mouse (*Epn1*fl/fl) was generated by homologous recombination at the *Epn1* locus with a targeting construct containing loxP sequence and a Frt-flanked neomycin cassette for selection

(removed by breeding with Flp mice, see also [*Pasula et al., 2012*]). *Epn1*$^{fl/fl}$ mice were interbred with *Epn2*$^{-/-}$ (*Chen et al., 2009*) and *Epn3*$^{-/-}$ (*Ko et al., 2010*) mice and with either 4-hydroxy-tamoxifen (OHT)-inducible Cre strain mice (*Badea et al., 2003*), expressing Cre recombinase under the control of the estrogen receptor promoter, or nestin-Cre mice (*Tronche et al., 1999*) to generate OHT-inducible and brain-specific epsin triple KO mice, respectively. All mice were of a C57BL/6J congenic background. Animal care and use was carried out in accordance with our institutional guidelines.

## Fibroblast cultures

Primary fibroblasts were isolated form conditional KO and WT mice from E18 embryos or P1 pups and cultured by standard methods. To achieve gene recombination and obtain epsin triple KO cells and their controls, cells were cultured for 7 days with addition of OHT (Sigma-Aldrich) to the culture medium at day 1 (3 µM) and day 4 (1 µM), a two-step 7-day long process. For cell counting, cells were plated at 50,000 cells per 35-mm dish. Two counts were performed for each of the three dishes of WT and epsin TKO cells using a hemocytometer, both at 1 and 3 days post-plating. The cell proliferation index represents the ratio between cell numbers at these two time points.

## Transfection

Plasmids were transfected by electroporation (Nucleofector, Amaxa, Cologne, Germany) for imaging experiments, or using Lipofectamine 2000 (Life Technologies) for immunoprecipitation experiments. siRNA oligos were transfected by using Lipofectamine RNAi MAX (Life Technologies), and cells were cultured for 48–72 hours before analysis. Double-stranded siRNAs were derived from the following references: Hip1R (mouse Hip1R, MMC.RNAI.N145070.12.1, and 12.2 from IDT, Coralville, IA, USA); control (NC1 negative control duplex from IDT).

## Immunofluorescence

Cells were plated and grown on 5 µg/ml human fibronectin (EMD-Millipore)-coated glass coverslips and fixed with 4% paraformaldehyde–4% sucrose in 0.1 M sodium phosphate buffer (pH 7.2) at room temperature. Coverslips were washed with 50 mM NH$_4$Cl pH 7.2, then blocked and permeabilized in 0.1% Triton X-100 and 3% bovine serum albumin in PBS. Primary and secondary antibody incubations were performed using the same buffer. Coverslips were finally rinsed in PBS and mounted in Prolong Gold + DAPI (Life Technologies). For surface staining, cells were incubated with an Alexa594-conjugated rat anti-HA antibody (Life Technologies) for 60 min on ice at 4°C. After extensive washing with cold PBS, cells were fixed as described above and counterstained with DAPI (Sigma-Aldrich) before coverslip mounting. Immunofluorescence data acquisition was performed using either a Zeiss Axioplan2 microscope (for the epifluorescence images) or a spinning disk confocal microscope (see below).

## Live imaging

For live imaging, cells were incubated at 37°C in the following buffer: 136 mM NaCl, 2.5 mM KCl, 2 mM CaCl$_2$, 1.3 mM MgCl$_2$, 3 mM D-glucose, and 10 mM Hepes-Na pH 7.4. Spinning-disk confocal microscopy was performed using the Improvision UltraVIEW VoX system (Perkin-Elmer, Walthman, MA, USA) built on a Nikon Ti-E inverted microscope, equipped with PlanApo objectives (60 × 1.45-NA) and controlled by Volocity (Improvision, Coventry, UK) software. Total internal reflection fluorescent (TIRF) microscopy was carried out on a Nikon TiE microscope equipped with 60 × 1.49-NA and 100 × 1.49-NA objectives. Excitation lights were provided by 488 nm (for GFP) and 561 nm (mCherry) diode-pumped solid-state lasers. An optical fiber coupled the lasers to the TIRF illuminator and an acousto-optic tunable filter controlled the output from the lasers. Fluorescent signals were detected with an EM-CCD camera (DU-887; Andor, Belfast, NIR) and acquisition was controlled by Andor iQ software. Images typically were sampled at 0.2 Hz with exposure times in the 4 to 6 s range.

## Transferrin uptake and transferrin receptor localization

For transferrin uptake, cells were electroporated with a plasmid encoding GFP-tagged human transferrin receptor (*Merrifield et al., 2005*). Cells were then starved for 2 hours with serum-free DMEM, incubated at 4°C for 30 min with 10 µg/ml Alexa594-conjugated human transferrin (Life Technologies), and then warmed up to 37°C for 15 min to allow internalization. Uptake was ended by a brief rinse with ice-cold PBS followed by fixation with 4% parafolmaldehyde–4% sucrose in 0.1 M sodium phosphate buffer pH 7.2 at room temperature. Cells were counterstained with DAPI to visualize nuclei.

## Cell surface protein biotinylation

Cells were rinsed with PBS and labeled on ice for 60 min with 1 mg/ml EZ-link Sulfo-NHS-SS-Biotin (Thermo Fisher, Walthman, MA, USA), rinsed on ice with 50 mM glycine in PBS pH 8.0, and then in PBS only. Cells were then lysed in modified RIPA buffer (1% Triton X-100, 0.1% SDS in 20 mM TRIS pH 7.5, 50 mM NaCl, EDTA 0.5 mM with protease and phosphatase inhibitors cocktails) and lysates were centrifuged at 16.200×g, at 4°C for 5 min. Biotinylated proteins were recovered on NeutrAvidin beads (Thermo Fisher). After washing with modified RIPA buffer, proteins were eluted with Laemmli SDS-PAGE sample buffer and boiling. Evaluation of protein levels in starting material, biotinylated (cell surface) and non-biotinylated (intracellular) fractions was assessed by Western blotting.

## Electron microscopy

Control and epsin TKO cells (~80% confluent) in 60-mm dishes were fixed in 2% glutaraldehyde–0.1 M sodium cacodylate. They were post-fixed with 1% $OsO_4$ in 1.5% $K_4Fe(CN)_6$ and 0.1 M sodium cacodylate, *en bloc* stained with 0.5% uranyl magnesium acetate, dehydrated and embedded in Embed 812. For morphometric analysis, cells were selected at a low magnification allowing confirmation that the entire outer perimeter was intact, but not visualizing clathrin-coated pits to make the selection process blind with respect to the phenotype of interest. Once selected, higher magnification images were taken around the periphery of the cell, and clathrin-coated structures within the categories defined in *Figure 2T* were counted all the way around the perimeter of each cell.

Electron microscopy reagents were purchased from Electron Microscopy Sciences (Hatfield, PA, USA).

## Immunoprecipitation

HeLa cells co-expressing FLAG-tagged epsin1, or FLAG alone and HA-tagged Syb2 or HA alone, were washed in ice-cold PBS and lysed in the lysis buffer (150 mM NaCl, 0.1% SDS, 0.5% Triton X-100, 10 mM EDTA, 20 mM Hepes pH 7.4, and protease inhibitor cocktail [Roche]). Cell lysates were then centrifuged at 16,000×g for 20 min at 4°C and supernatants were incubated, under rotation, with agarose-conjugated anti-HA beads (Roche) for 1 hour at 4°C. Beads were then extensively washed in cold lysis buffer and bound proteins were eluted in Laemmli sample buffer and boiled for 5 min.

## Mouse brain cytosol preparation

4 weeks-old WT, epsin TKO or Hip1/Hip1R double KO mouse brains were homogenized in homogenization buffer (25 mM Tris pH 8.0, 500 mM KCl, 250 mM sucrose, 2 mM EGTA, 1 mM dithiothreitol) in the presence of a protease inhibitor cocktail (Roche). The lysate was centrifuged at 160,000×g for 2 hours in a TL100.2 rotor (Beckman Coulter, Indianapolis, IN, USA). The resulting supernatant was buffer-exchanged on PD-10 columns (GE Healthcare) at room temperature into cytosolic buffer (25 mM Hepes pH 7.4, 120 mM potassium glutamate, 20 mM potassium chloride, 2.5 mM magnesium acetate, 5 mM EGTA). Protease inhibitors were added and aliquots of cytosol were immediately frozen in liquid nitrogen and stored at −80°C for up to 2 months.

## Plasma membrane sheets preparation

PTK2 cells stably expressing PM–GFP (*Chen and De Camilli, 2005*) were cultured at 37°C in 10% $CO_2$ in minimum essential Eagle medium (Life Technologies) supplemented with 10% (vol/vol) fetal bovine serum, 100 μg/ml penicillin/streptomycin, and 0.5 mg/ml G418. Cells were grown for 24–48 hours until confluent in poly-d-lysine coated MatTek dishes (MatTek, Ashland, MA, USA). These dishes were prepared by exposing them to 20 μg/ml poly-d-lysine (Sigma-Aldrich) for 30 min and washed overnight. To prepare plasma membrane sheets, cells were rinsed with PBS and then sheared in ice-cold cytosolic buffer by a brief pulse of sonication (about 0.5 s) using a cell disruptor (VirTis Ultrasonics) set at 20% of output power with a 1/8-inch microprobe positioned at about 15 mm above the dish. Membrane sheets were rinsed in cytosolic buffer and used for the in vitro assay within 20 min.

Generation of endocytic budding intermediates from plasma membrane sheets was achieved by incubating them at 37°C with cytosol in the presence of nucleotides in the following concentrations: 2 mg/ml cytosol, 1.5 mM ATP and 150 μM GTPγS. Samples were supplemented with an ATP-regenerating system consisting of 16.7 mM creatine phosphate and 16.7 U/ml creatine phosphokinase. The reaction was stopped by washing with cytosolic buffer and fixing samples with 4% paraformaldehyde–4% sucrose in 0.1 M sodium phosphate buffer pH 7.2 for 15 min at room temperature.

## Actin co-sedimentation assay

250 µg of rabbit muscle G-actin (Sigma-Aldrich) were polymerized for 30 min at 25°C in polymerization buffer (50 mM KCl, 2 mM MgCl$_2$, 1 mM ATP). Recombinant GST-epsin fragments (final concentration 5 µM) were pre-cleared by centrifugation at 150,000×$g$ for 1 hour at 4°C and incubated with F-actin (final concentration: 15 µM). After 30 min of incubation at room temperature, samples were centrifuged at 150,000×$g$ for 1.5 hours at 4°C. Supernatants and pellets were brought to equal volumes of SDS/PAGE Laemmli buffer, and samples were analyzed by SDS/PAGE using 4–12% Bis-Tris NuPAGE gel in MES buffer (Life Technologies) followed by anti-GST immunoblotting.

## Recombinant protein purification

For GST-fusion protein production, BL21-DE3 *E. coli* cells were transformed by heat-shock at 42°C with pGEX-4T alone or pGEX-4T constructs. Large-scale cultures were grown to log phase (about 3 hours) in Luria-Bertani broth containing ampicillin (100 mg/ml) at 37°C, under shaking at 200–250 rpm. Cultures were then induced with 500 mM isopropyl-β-D-1-thiogalactopyranoside (IPTG) at 37°C for 3 hours and pelleted by centrifugation at 3000×$g$ for 15 min at 4°C. Bacteria were lysed in 150 mM NaCl, 4 mM DTT, 10 mM Hepes pH 7.4, and protease inhibitor cocktail (Roche) in the presence of 1% Triton X-100. Lysates were cleared by centrifugation at 22,000×$g$ for 20 min at 4°C, and the resulting supernatants were incubated with Glutathione-Sepharose 4 Fast Flow beads (GE Healthcare) for 1 hour at 4°C. After thorough washings with ice-cold buffer (without Triton X-100), proteins were eluted by incubating the beads with 30 mM glutathione, 150 mM NaCl, 4 mM DTT, and 10 mM HEPES pH 8.

For His-tagged protein generation, BL21-DE3 pLys-S *E coli* were transformed as described above with a pET21a (+) encoding a 6xHis-tagged fusion protein. After reaching the log phase at 37°C, bacteria were induced overnight at 18°C with 100 mM IPTG. Bacterial pellet was lysed in buffer (150 mM NaCl, 1 mM β-mercaptoethanol, 20 mM HEPES pH 7.4, 1% Triton X-100) containing 15 mM imidazole, and the cleared supernatants were incubated with Ni$^{2+}$–NTA agarose beads for 1 hour at 4°C. Samples were washed in the same buffer, but containing 30 mM imidazole, and finally recombinant proteins were eluted in the same buffer containing 300 mM imidazole.

Both GST- and His-tagged proteins were subsequently purified by gel filtration using a Superdex S200 or a Superdex S75 column (GE Healthcare) on the Akta Pure FPLC system (GE Healthcare). Finally, proteins were buffer exchanged in 150 mM NaCl, 4 mM DTT, 10 mM Hepes pH 7.4 by High-Prep 26/10 Desalting Column (GE Healthcare). Protein purity was assessed by Coomassie Blue staining of samples run on 4–20% gradient Tris/Glycine or 16.5% Tris-Tricine MiniProtean gels (Biorad, Hercules, CA, USA).

## GST pull-down assays to assess protein–protein interactions

GST-tagged bait proteins bound to glutathione beads were incubated overnight at 4°C with constant agitation with 6xHis-tagged prey proteins in 150 mM NaCl, 4 mM DTT, 4 mM β-mercaptoethanol, 0.1% NP-40, 0.05% BSA, 20 mM HEPES pH 7.4. Reaction mixtures were subsequently cleared by centrifugation at 3000×$g$ for 5 min at 4°C; supernatants were removed and beads were washed three times with ice-cold buffer. Bound proteins were eluted in Laemmli buffer and analyzed by separation on 16.5% Tris-Tricine gradient Mini-Protean gels (Biorad) followed by Western blotting with anti-His antibody. In experiments where the effect of phosphoinositol groups was tested, the incubation mixture was supplemented with 50 µM final of diC8PI(4,5)P$_2$, IP$_3$(1,4,5) (Avanti Polar Lipids, Alabaster, AL, USA) or IP$_6$(1,4,5) (phytic acid dipotassium salt, Sigma-Aldrich).

## Generation of Syb2–HA stable cell line

Mouse fibroblasts stably expressing Syb2–HA were generated by transducing WT and conditional epsin double KO mouse fibroblasts with a Syb2–HA encoding retrovirus. The virus was produced in the Phoenix helper-free retrovirus producer line (293T cell line transformed with adenovirus E1a and carrying a temperature sensitive T antigen co-selected with neomycin). Briefly 10 µg of pBABE puro-Syb2–HA were transfected by Lipofectamine 2000 (Life Technologies) into Phoenix cells. In parallel WT and conditional epsin double KO fibroblasts were plated. 24 hours after transfection, the medium from the Phoenix cell dishes was removed, supplemented with 8 µg/ml polybrene (cationic polymer used to increase the retroviral infection efficiency in cell culture) and filtered to remove cell debris. The filtrate was added to the mouse fibroblasts and removed after 4–6 hours incubation at 37°C. 48 hours after the initial viral incubation fibroblasts were splitted in Dulbecco's

Modified Eagle Medium (Life Technologies). pBABE puro vector carries a puromycin resistance gene that was used to select the Syb2–HA expressing cells by supplementing the culture medium with 4 µg/ml puromicyn ([*Ferguson et al., 2009*] ,Sigma-Aldrich).

## Quantification and statistical analysis

Fluorescent signals were quantified with Fiji ImageJ (http://fiji.sc/wiki/index.php/Fiji) or the Volocity 3D Image Analysis software (Improvision). Immunoblots were analyzed by Fiji ImageJ or Image Studio (Licor Bioscience, Lincoln, NE, USA). Graphical presentations were made using Graph Pad Prism (Graph Pad Software, La Jolla, CA, USA).

Statistical analyses were performed by Graph Pad Prism using Student's *t* test for independent samples or one-way Anova.

## Miscellaneous procedures

Bicinchoninic acid protein quantification (Thermo Fisher), SDS-PAGE electrophoresis, immunoblotting, and GST-pulldowns from brain homogenates were performed by standard procedures (*Ferguson et al., 2009*).

## Acknowledgements

We thank Shawn Ferguson (Yale University) and Yixian Zheng (Carnegie Institution for Science, Baltimore, MD) for discussion and suggestions; Frank Wilson, Lijuan Liu, and Louise Lucast for superb technical assistance and Alanna Coughran and Abigail Soyombo for help with the Hip1/Hip1R DKO mice. Generous gifts of key reagents are acknowledged in the Materials and methods section. This work was supported in part by grants R37NS036251, DK082700, DK45735 and DA018343 to PDC, R01GM60979 to BW, R01CA098730 to TR and a Grant from the American Heart Association (#0835544N) to HC. RF-B was the recipient of a Feodor Lynen postdoctoral fellowship from the Alexander von Humboldt Foundation, EWS was supported by NIH 5T32GM007223-38/39 and KW by NIGMST32007231.

## Additional information

### Funding

| Funder | Grant reference number | Author |
| --- | --- | --- |
| Howard Hughes Medical Institute | | Mirko Messa, Rubén Fernández-Busnadiego, Elizabeth Wen Sun, Hong Chen, Heather Czapla, Yumei Wu, Genevieve Ko, Pietro De Camilli |
| National Institute of Neurological Disorders and Stroke | R37NS036251 | Pietro De Camilli |
| National Institute on Drug Abuse | DA018343 | Pietro De Camilli |
| National Institute of Diabetes and Digestive and Kidney Diseases | DK082700, DK45735 | Pietro De Camilli |
| National Institute of General Medical Sciences | R01GM60979 | Beverly Wendland |
| National Cancer Institute | R01CA098730 | Theodora Ross |
| American Heart Association | 0835544N | Hong Chen |
| Alexander von Humboldt-Stiftung | | Rubén Fernández-Busnadiego |
| National Institute of General Medical Sciences | NIGMST32007231 | Kristie Wrasman |
| National Institutes of Health | 5T32GM007223-38/39 | Elizabeth Wen Sun |

The funders had no role in study design, data collection and interpretation, or the decision to submit the work for publication.

## Author contributions

MM, RF-B, PDC, Conception and design, Acquisition of data, Analysis and interpretation of data, Drafting or revising the article, Contributed unpublished essential data or reagents; EWS, Acquisition of data, Analysis and interpretation of data, Drafting or revising the article; HC, TR, Drafting or revising the article, Contributed unpublished essential data or reagents; HC, YW, Acquisition of data, Analysis and interpretation of data; KW, Conception and design, Acquisition of data, Analysis and interpretation of data; GK, Acquisition of data, Contributed unpublished essential data or reagents; BW, Conception and design, Analysis and interpretation of data, Drafting or revising the article, Contributed unpublished essential data or reagents

## Ethics

Animal experimentation: The institutional animal care and use committee (IACUC) of the Yale University and the approved animal protocol is 2012-07422. The institutional guidelines for the care and use of laboratory animals were followed.

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
