## [Decision Letter]

Thank you for sending your work entitled “Epsin deficiency impairs endocytosis by stalling the actin-dependent invagination of endocytic clathrin coated pits” for consideration at *eLife.* Your article has been favorably evaluated by Randy Schekman (Senior editor) and 2 reviewers, one of whom is a member of our Board of Reviewing Editors. After sending the original decision to you, we received a further peer review from reviewer 3. We realize this is unusual but ask that you try to address these issues in your revision; because the decision was sent, it should not require additional experimental work beyond the original review but if something is easily checked that would be great.

The Reviewing editor and the other reviewers discussed their comments before we reached this decision, and the Reviewing editor has assembled the following comments to help you prepare a revised submission.

This manuscript makes use of epsin triple knockout fibroblasts, to try to clear up some of the confusion surrounding the function(s) of the epsin family of proteins. Although it is widely agreed that epsins are important players in clathrin-mediated endocytosis, there is still a lot of controversy over what they actually do. Part of the problem is that there are three epsin isoforms in mammals, encoded by different genes. Worms and flies both have only a single epsin gene and knockouts have been shown to impair Notch signaling (which requires clathrin-mediated endocytosis). Similarly, in a study by the De Camilli lab published five years ago, knocking out both of the ubiquitously expressed mouse epsin genes, epsin 1 and epsin 2, causes embryonic lethality in a way that can be explained by impairment of multiple Notch pathways. Importantly, clathrin-mediated endocytosis in general is not abolished in any of the epsin knockout animals.

However, there was a high profile paper from the McMahon lab, published two years ago in Cell, claiming that if all three epsin genes were knocked down by siRNA in HeLa, BSC1, or RPE1 cells, clathrin-mediated endocytosis was effectively abolished: the phenotype was as severe as when clathrin or AP-2 were knocked down. The authors claimed that the reason this phenotype hadn't been observed before was that the three epsin isoforms were functionally redundant. This is in spite of the fact that expression of epsin 3 is highly tissue-specific, and the authors never investigated whether it was actually expressed in these cell lines. The Boucrot et al. paper also reported that in the epsin triple knockdown, the coated pits were arrested at scission, and that epsin could substitute for dynamin in dynamin-depleted cells. Micrographs showed “multiheaded” clathrin-coated structures, appearing like bunches of grapes. It was difficult to know what to make of this paper, but until the present study, no other piece of work has come along to challenge it.

The present manuscript is timely, and deserves to be published where it will be highly visible. It represents a huge amount of work, and presents very high quality data. The take-home message is that in the epsin triple knockout cells, there is a partial block in clathrin-mediated endocytosis, but the phenotype is quite different from that reported by Boucrot et al. Most importantly, the authors present several lines of evidence indicating that epsin plays a key role in linking clathrin to the actin cytoskeleton. This is the first evidence for such a role for epsin, and for a functional association between Hip1R and epsin, in animal cells; however, it fits in nicely with studies that have been carried out in yeast and *Dictyostelium*. The authors should be able to address all of the comments by making amendments to the text.

1) It would be helpful to know why the authors had to use a conditional knockout for epsin 1. Their strategy was to generate knockout mice with no epsin 2 or epsin 3, and then to cross them with a floxed epsin 1 strain, to generate the strain Epn1loxP/loxP; Epn2-/-; Enp3-/-. I assume that the authors initially tried to cross their Epn2-/-; Enp3-/- mice, which are healthy, with their Epn1-/- mice, which are also healthy, but that the triply homozygous embryos died at an early stage; however, it would be good if this could be spelled out. The authors then crossed their mice with an inducible Cre recombinase strain and isolated embryonic fibroblasts. The cells were typically treated for 7 days with a drug to induce Cre expression, thereby excising the epsin 1 gene. It is clear from the Western blots in Figure 1 that even after 7 days, trace amounts of epsin 1 are still detectable. Is this because the gene had not been excised in 100% of the cells, or because “old” protein had not been completely degraded? The immunofluorescence images, in which epsin 1 is undetectable in the drug-treated cells, suggest the former. The authors do not say why they didn't treat the cells even longer with the drug, or try to isolate a clonal population completely lacking all three genes, but I assume it was because the cells were no longer viable? Again, it would be helpful if this could be clarified.

2) To what extent was clathrin-mediated endocytosis impaired in the triple knockout cells? The Boucrot et al. paper claims that they got a complete block, but the data in Figure 3 suggest something more modest (e.g., 50% or so). Please clarify this.

3) Based on their immunofluorescence labeling, the authors report a “robust increase in the density of clathrin-coated pits”, which they also found to be more clustered (Figure 2). However, quantification of their EM data (Figure 2) does not reflect this difference: if one adds up the data for all of the coated structures, the numbers are remarkably similar: ∼0.095 coated structures per micron cell perimeter in the control, and ∼0.1 in the triple knockout. Is this because individual pits are wider, or because clusters are treated as single structures? It is interesting to note that a similar phenotype was reported for CALM knockdown cells (Meyerholz et al., 2005, Traffic 6: 1225-1234). Could this be due to a similar phenomenon?

4) There is some confusion in the paper about the relationship between epsin and epsinR, and about the function of epsinR. In opisthokonts (i.e., animals and fungi), there are two families of genes with similar ENTH (epsin N-terminal homology) domains. Epsin genes (e.g., EPN1, EPN2, and EPN3 in humans, and ENT1 and ENT2 in yeast) encode proteins that are associated with AP-2 and clathrin at the plasma membrane, and that contain ubiquitin-interacting motifs (UIMs). EpsinR genes (e.g., CLINT1 in humans, and ENT3 and ENT5 in yeast) encode proteins that are associated with AP-1 and clathrin at intracellular membranes, and that interact with the Q-SNARE vti1b/Vti1p. However, most other eukaryotes have only one such protein, which has been defined as epsinR because it lacks UIMs (e.g., see Gabernet-Castello et al., 2009, Traffic 10: 894-911). Thus, when the authors report that their studies are in line with studies on Dictyostelium, where links between epsin, Hip1R, and actin were also reported, they need to be aware that the Dictyostelium protein is thought to be more like epsinR. Interestingly, however, the Dictyostelium protein localizes to both the plasma membrane and intracellular membranes, suggesting that the ancestral epsin/epsinR was able to contribute to both clathrin-mediated endocytosis and clathrin-mediated intracellular trafficking, but that then after a gene duplication in pre-opisthokonts, the two descendants became specialized for different pathways. Again, it would be good if there could be some clarification about epsin vs. epsinR in non-opisthokonts. In addition, the authors need to correct their statement on page 17 that “epsin 1R” binds V-SNAREs, because the correct name for the protein is epsinR, and the SNARE that it binds is a T-SNARE, not a V-SNARE.

5) The structures in Figure 7 are so tiny that it is impossible to see the epsin, let alone what it does.

6) Please be sure to discuss clearly why these results differ from those reported previously, as summarized in this review above.

Reviewer #3

1) The previously published data from yeast and the new data in this paper imply that Epsin and Hip1R may stabilize each other at endocytic sites. If the authors could address the mechanism of this stabilization, it would greatly improve the significance and impact of the work; at minimum, please discuss.

2) The focus of the paper (reflected in the title) is on Epsin functions in endocytosis. However, the paper starts with the mitotic defects in Epsin TKO cells. This may distract the readers' attention. Figure 1 might be better in the supplemental results.

3) Figure 2, Clathrin localization should be probed by two-color imaging together with dynamin, or endophilin, or Myosin IE in the TKO cells. This should not be difficult.

4) Based on the results presented in Figure 4, one of the Epsin functions is to recruit Hip1R to endocytic sites. However, it is not clear that the phenotype shown in the TKO cells is due to loss of HipR1 function at endocytic sites, or if Epsin shares a similar function with HIP1R, or both. Thus, can the authors compare the phenotypes between TKO cells and Hip1R knockout cells?

5) The cell free system could be useful in many cases, however, it did not provide more in the way of novel insights to this study. In fact, the Myosin 1E localization shown in Figure 5 using the cell free system is inconsistent with the data shown in Figure 2 in fixed cells. For the F-actin binding experiments, the protein size of ENTH domain seems to be the same as actin, so how the authors can conclude that the ENTH domain does not bind to F-actin is not clear. The bands are not labeled. Please clarify.

6) The model in Figure 7 is too general and is not useful in terms of highlighting the main points of the paper. Please improve and show epsin clearly.

---

## [Author Response]

*1) It would be helpful to know why the authors had to use a conditional knockout for epsin 1. Their strategy was to generate knockout mice with no epsin 2 or epsin 3, and then to cross them with a floxed epsin 1 strain, to generate the strain Epn1loxP/loxP; Epn2-/-; Enp3-/-. I assume that the authors initially tried to cross their Epn2-/-; Enp3-/- mice, which are healthy, with their Epn1-/- mice, which are also healthy, but that the triply homozygous embryos died at an early stage; however, it would be good if this could be spelled out*.

Correct, we did so as the animals die at an early stage; we have now made this clearer in the text. We now state:

“As the germline deletion of even only two Epn genes results in embryonic lethality, a conditional approach was used to generate Epn1, Epn2 and Epn3 triple KO cells.”

*The authors then crossed their mice with an inducible Cre recombinase strain and isolated embryonic fibroblasts. The cells were typically treated for 7 days with a drug to induce Cre expression, thereby excising the epsin 1 gene. It is clear from the Western blots in*
Figure 1
*that even after 7 days, trace amounts of epsin 1 are still detectable. Is this because the gene had not been excised in 100% of the cells, or because “old” protein had not been completely degraded? The immunofluorescence images, in which epsin 1 is undetectable in the drug-treated cells, suggest the former. The authors do not say why they didn't treat the cells even longer with the drug, or try to isolate a clonal population completely lacking all three genes, but I assume it was because the cells were no longer viable? Again, it would be helpful if this could be clarified*.

We do find after the 7-day treatment a few cells where epsin immunoreactivity has not completely disappeared. Thus, we believe that the residual minimal amount of epsin is explained by a combination of both factors raised by the reviewer. We now state:

“The extremely small amount of residual epsin 1 was likely explained by delayed gene recombination in a few cells, where residual epsin 1 immunoreactivity, much lower than in controls, was observed by immunofluorescence.”

We did not treat cells for a longer time with the drug, as a longer treatment resulted in cell lethality.

*2) To what extent was clathrin-mediated endocytosis impaired in the triple knockout cells? The Boucrot et al. paper claims that they got a complete block, but the data in*
Figure 3
*suggest something more modest (e.g., 50% or so). Please clarify this*.

The reviewers probably refer to Figure 2, as this is the figure (not Figure 3) that describes the defect in clathrin-mediated endocytosis. The robustness of this defect is clearly demonstrated by the striking difference in the localization of transferrin (after a 15 min incubation, Figure 2) and of the transferrin receptor (a classical marker of clathrin-mediated endocytosis) as assessed by immunofluorescence (Figure 2) and surface biotinylation (Figure 2). In WT cells, the bulk of the receptor, as well as of transferrin after 15 min incubation, are inside the cell, while in triple KO cells the overwhelming majority of the receptor, as well as of transferrin, are at the cell surface. This effect is comparable to the one we had observed in dynamin null cells ([26], PMID:20059951, Figure 1).

We do appreciate that the analysis of transferrin fluorescence reported in the original figure (original Figure 2, which showed the ratio between surface and internalized fluorescent signal as detected by confocal microscopy) did not quantify with accurate precision the internalization and most likely underestimated the internalization defect. The fluorescence images that we present (Figure 2), along with the biochemical biotinylation data (Figure 2), strongly support a massive internalization defect, consistent with the Boucrot et al. paper, in which the analysis of transferrin internalization was carried out by FACS in epsin triple knockdown cells. For this reason, we did not deem it necessary to repeat the analysis of transferrin internalization using a biochemical assay. However, we could do such an assay, if the reviewer considers this essential towards to acceptance of our manuscript.

*3) Based on their immunofluorescence labeling, the authors report a “robust increase in the density of clathrin-coated pits”, which they also found to be more clustered (*Figure 2*). However, quantification of their EM data (*Figure 2*) does not reflect this difference: if one adds up the data for all of the coated structures, the numbers are remarkably similar: ∼0.095 coated structures per micron cell perimeter in the control, and ∼0.1 in the triple knockout. Is this because individual pits are wider, or because clusters are treated as single structures?*

We thank the reviewers for this comment; we realize the EM quantification was not reflecting the increase in clathrin-coated pits density observed by IF. In response to this comment we have reassessed the analysis on a higher number of samples for both the control and the epsin TKO. We found the total number of coated structures is significantly different between control and TKO and we show it in the new Figure 2.

It is interesting to note that a similar phenotype was reported for CALM knockdown cells (Meyerholz et al., 2005, Traffic 6: 1225-1234). Could this be due to a similar phenomenon?

It is really difficult to say if this is a mechanistically similar phenomenon and we prefer not to speculate on this point.

*4) There is some confusion in the paper about the relationship between epsin and epsinR, and about the function of epsinR. In opisthokonts (i.e., animals and fungi), there are two families of genes with similar ENTH (epsin N-terminal homology) domains. Epsin genes (e.g., EPN1, EPN2, and EPN3 in humans, and ENT1 and ENT2 in yeast) encode proteins that are associated with AP-2 and clathrin at the plasma membrane, and that contain ubiquitin-interacting motifs (UIMs). EpsinR genes (e.g., CLINT1 in humans, and ENT3 and ENT5 in yeast) encode proteins that are associated with AP-1 and clathrin at intracellular membranes, and that interact with the Q-SNARE vti1b/Vti1p. However, most other eukaryotes have only one such protein, which has been defined as epsinR because it lacks UIMs (e.g., see Gabernet-Castello et al., 2009, Traffic 10: 894-911). Thus, when the authors report that their studies are in line with studies on Dictyostelium, where links between epsin, Hip1R, and actin were also reported, they need to be aware that the Dictyostelium protein is thought to be more like epsinR. Interestingly, however, the Dictyostelium protein localizes to both the plasma membrane and intracellular membranes, suggesting that the ancestral epsin/epsinR was able to contribute to both clathrin-mediated endocytosis and clathrin-mediated intracellular trafficking, but that then after a gene duplication in pre-opisthokonts, the two descendants became specialized for different pathways. Again, it would be good if there could be some clarification about epsin vs. epsinR in non-opisthokonts*.

We thank the reviewers for raising this point. We have now addressed it with the following text:

“It may additionally participate in the recruitment to the pit of ubiquitinated cargo via its UIMs, a function that does not apply to Dictyostelium epsin, which expresses a single epsin family protein lacking UIMs.”

We prefer, however, not to dwell into the discussion of opisthokonts versus amoebas, as it is really peripheral to the main topic of the paper.

*In addition, the authors need to correct their statement on page 17 that “epsin 1R” binds V-SNAREs, because the correct name for the protein is epsinR, and the SNARE that it binds is a T-SNARE, not a V-SNARE*.

The reviewers are correct. This comment made us reconsider the terminology used and we decided to simply speak of SNAREs.

*5) The structures in*
Figure 7
*are so tiny that it is impossible to see the epsin, let alone what it does*.

The point of Figure 7 was not to show mechanistic details about epsin function but to show its sites of action. However, in response to this comment we have added two fields demonstrating mechanistic details.

*6) Please be sure to discuss clearly why these results differ from those reported previously, as summarized in this review above*.

We have introduced the following sentence at the beginning of the second paragraph of the Discussion:

“A main conclusion of our study is the occurrence of a close coupling between the function of epsin and the dynamics of the actin cytoskeleton. While such coupling had been suggested by studies in unicellular organisms, studies of epsin in cells of metazoan had emphasized its role as a bilayer deforming protein and as a cargo-specific adaptor.”

Concerning the discrepancy from the Boucrot et al. paper, we believe to have addressed it with the following statement already present in the original manuscript:

“Recently it was proposed that a main role of epsin in endocytic clathrin-coated pit dynamics is to help mediate membrane fission via the membrane remodeling properties of the amphipathic helix zero of its ENTH domain. While this action seems plausible, our study strongly suggests that epsin becomes critically important at earlier stages of clathrin coat maturation, as we have observed an accumulation of shallow and U-shaped endocytic clathrin-coated pits in epsin TKO cells. The diffuse, rather than punctate distribution of proteins that assemble at the necks of deeply invaginated endocytic clathrin-coated pits, such as dynamin 2, endophilin 2 and myosin 1E further supports a stalling of pits at an early stage in epsin TKO cells. In contrast to observations made in epsin triple KD cells we have not observed clearly multiheaded pits, although the dome of some pits was irregular. Multiheaded pits observed in epsin triple KD cells had been interpreted as reflecting a defect in fission. However, they do not have the constricted neck of multiheaded pits observed in dynamin mutant cells and at least some of them could represent a clustering of shallow or U-shaped pits. We note that the multiple endocytic functions of epsin are consistent with its localization throughout the coat, rather than a selective localization at the bud neck to mediate fission.”

In our opinion, this paragraph accurately addresses the discrepancy with the paper by Boucrot et al. (McMahon’s lab).

Reviewer #3

*1) The previously published data from yeast and the new data in this paper imply that Epsin and Hip1R may stabilize each other at endocytic sites. If the authors could address the mechanism of this stabilization, it would greatly improve the significance and impact of the work; at minimum, please discuss*.

Based on our data we conclude that in mammalian cells epsin plays a major role in the recruitment of Hip1R to endocytic pits, but not vice versa. We note that a role of epsin in the recruitment of Hip1R was also observed in Dictyostelium ([10], PMID: 20923836; quoted in the manuscript), while in yeast, it is Hip1R (Sla2) that recruits epsin (Ent1 and 2, [58], PMID:22927393; quoted in the manuscript). Thus, while the partnership between epsin and Hip1R is evolutionary conserved, the relative importance of epsin and Hip1R in the recruitment of the other protein to coated pits varies in different species.

*2) The focus of the paper (reflected in the title) is on Epsin functions in endocytosis. However, the paper starts with the mitotic defects in Epsin TKO cells. This may distract the readers' attention.*
Figure 1
*might be better in the supplemental results*.

We prefer to leave the discussion about the mitotic defect in Figure 1, as this defect has a major impact on the morphology of epsin triple KO cells. The reader would be puzzled by the huge size of triple KO cells shown in subsequent figures (Figure 2 and Figure 3) if this defect was not discussed early on in the Results. In any case, as it is now clear that many endocytic proteins, including clathrin, also play an important role in mitosis, the cell division defect of epsin triple KO cells is not totally unrelated to the endocytic function of epsin.

*3)*
Figure 2*, Clathrin localization should be probed by two-color imaging together with dynamin, or endophilin, or Myosin IE in the TKO cells. This should not be difficult*.

We have done this and we show these images in revised Figure 2. We have also replaced two figures with better images: former Figure 3, where two of the fields had been erroneously switched, and former Figure 3.

*4) Based on the results presented in*
Figure 4*, one of the Epsin functions is to recruit Hip1R to endocytic sites. However, it is not clear that the phenotype shown in the TKO cells is due to loss of HipR1 function at endocytic sites, or if Epsin shares a similar function with HIP1R, or both. Thus, can the authors compare the phenotypes between TKO cells and Hip1R knockout cells?*

While the deletion of even only two epsins results in early embryonic lethality ([18], PMID: 19666558; quoted in the manuscript), Hip1 and Hip1R double KO mice live to adulthood, although they have multiple defects. Accordingly, it was reported:

“Hip1 and Hip1R are not necessary for endocytosis, but are necessary for the maintenance of diverse adult tissues in vivo” ([9], PMID: 17452370; quoted in the manuscript). In view of these previous studies, and of the dramatic endocytic defect produced by the absence of all three epsins, we conclude that the phenotype observed in epsin triple KO cells is not simply due to the loss of HIP1/HIP1R function.

*5) The cell free system could be useful in many cases, however, it did not provide more in the way of novel insights to this study. In fact, the Myosin 1E localization shown in*
Figure 5
*using the cell free system is inconsistent with the data shown in*
Figure 2
*in fixed cells*.

We consider the cell free system an important complement to the *in situ* studies. Results in this system generally confirm what we have found in intact cells. While the reviewer is correct in pointing out some discrepancy between the localization of myosin 1E in the cell free system and in cells (i.e. two very different experimental conditions), both assays concur in demonstrating a mislocalization of the protein.

*For the F-actin binding experiments, the protein size of ENTH domain seems to be the same as actin, so how the authors can conclude that the ENTH domain does not bind to F-actin is not clear. The bands are not labeled. Please clarify*.

The reviewer is correct in stating that in our original figure (Figure 5) the lack of binding of the ENTH domain of epsin to actin could not be appropriately assessed in the Coomassie blue stained gel because of the co-migration of the ENTH domain (GST-ENTH) with actin. To overcome this problem, the original figure has been replaced with a new figure (new Figure 5) in which the GST fusions of the epsin fragments are revealed by anti-GST western blots rather than by Coomassie blue staining.

*6) The model in*
Figure 7
*is too general and is not useful in terms of highlighting the main points of the paper. Please improve and show epsin clearly*.

As discussed above, the point of Figure 7 was not to show mechanistic details about epsin function but to show its sites of action. However, in response to this comment we have added two fields demonstrating mechanistic details.